JCB Journal of Cell Biology

# ANKRD24 organizes TRIOBP to reinforce stereocilia insertion points

Jocelyn F. Krey[1,2] , Chang Liu[3], Inna A. Belyantseva[4] , Michael Bateschell[1,2], Rachel A. Dumont[1,2] , Jennifer Goldsmith[1,2] , Paroma Chatterjee[1,2], Rachel S. Morrill[1,2] , Lev M. Fedorov[5], Sarah Foster[1], Jinkyung Kim[6], Alfred L. Nuttall[1], Sherri M. Jones[7] , Dongseok Choi[8], Thomas B. Friedman[4] , Anthony J. Ricci[6] , Bo Zhao[3] , and Peter G. Barr-Gillespie[1,2]

The stereocilia rootlet is a key structure in vertebrate hair cells, anchoring stereocilia firmly into the cell's cuticular plate and protecting them from overstimulation. Using superresolution microscopy, we show that the ankyrin-repeat protein ANKRD24 concentrates at the stereocilia insertion point, forming a ring at the junction between the lower and upper rootlets. Annular ANKRD24 continues into the lower rootlet, where it surrounds and binds TRIOBP-5, which itself bundles rootlet F-actin. TRIOBP-5 is mislocalized in *Ankrd24[KO/KO]* hair cells, and ANKRD24 no longer localizes with rootlets in mice lacking TRIOBP-5; exogenous DsRed–TRIOBP-5 restores endogenous ANKRD24 to rootlets in these mice. *Ankrd24[KO/KO]* mice show progressive hearing loss and diminished recovery of auditory function after noise damage, as well as increased susceptibility to overstimulation of the hair bundle. We propose that ANKRD24 bridges the apical plasma membrane with the lower rootlet, maintaining a normal distribution of TRIOBP-5. Together with TRIOBP-5, ANKRD24 organizes rootlets to enable hearing with long-term resilience.

## Introduction

Hair cells of the inner ear transduce mechanical forces arising from sound stimuli and movements of the head using hair bundles, each containing ~100 stereocilia (Fettiplace, 2017). Because they are composed of hundreds of unidirectionally oriented and tightly packed actin filaments (F-actin), stereocilia are very stiff along their lengths. At their insertion into the apical surface of the cell, however, each stereocilium has only a few dozen F-actin filaments, allowing it to pivot easily (Howard and Ashmore, 1986).

Stereocilia rootlets traverse this pivot point. Rootlets are composed of bundled actin filaments that are spaced considerably closer than those of the stereocilium shaft (Itoh and Nakashima, 1980; Tilney et al., 1980; Itoh, 1982; Slepecky and Chamberlain, 1982; Song et al., 2020). The upper part of the rootlet extends into a stereocilium, and the lower rootlet extends into the cuticular plate, a meshwork of actin filaments just below the apical surface of a hair cell (DeRosier and Tilney, 1989; Furness et al., 2008). Rootlets are damaged by loud sound (Liberman, 1987), which suggests that they bear large forces during overstimulation. Rootlets are proposed to reinforce the insertion points of stereocilia to increase pivot durability

(Kitajiri et al., 2010) and minimize damage to stereocilia (Furness et al., 2008; Pacentine et al., 2020). TRIOBP is a major component of the rootlet, wrapping around the actin core; actin filaments bundled in vitro with TRIOBP-4 have filament spacing similar to those in rootlets (Kitajiri et al., 2010). Moreover, rootlets do not develop in mice lacking TRIOBP-4 and TRIOBP-5, and these mice have unusually compliant hair bundles (Kitajiri et al., 2010; Katsuno et al., 2019). In humans, loss of *TRIOBP* underlies the deafness DFNB28 (Riazuddin et al., 2006; Shahin et al., 2006).

Here we have identified a novel component of rootlets, ANKRD24, which belongs to the N-Ank family (Wolf et al., 2019). ANKRD24 concentrates in the form of a ring at the apical membrane, where the stereocilia insert into the cell's apical surface. ANKRD24 contains not only a membrane-binding N-terminal domain and ankyrin repeats, but also two coiled-coil domains, which mediate ANKRD24 self-association and interaction with TRIOBP. Although TRIOBP-5 recruits ANKRD24 to rootlets, if ANKRD24 is absent, TRIOBP-5 shifts away from the insertion point and is less uniformly distributed among rootlets. ANKRD24 thus maintains an even allocation of TRIOBP-5 in row

[1]Oregon Hearing Research Center, Oregon Health & Science University, Portland, OR; [2]Vollum Institute, Oregon Health & Science University, Portland, OR; [3]Department of Otolaryngology—Head and Neck Surgery, Indiana University School of Medicine, Indianapolis, IN; [4]Laboratory of Molecular Genetics, National Institute on Deafness and Other Communication Disorders, National Institutes of Health, Bethesda, MD; [5]Transgenic Mouse Models, University Shared Resources Program, Oregon Health & Science University, Portland, OR; [6]Department of Otolaryngology—Head & Neck Surgery, Stanford University, Stanford, CA; [7]Department of Special Education and Communication Disorders, University of Nebraska-Lincoln, Lincoln, NE; [8]OHSU-PSU School of Public Health, Oregon Health & Science University, Portland, OR.

Correspondence to Peter G. Barr-Gillespie: gillespp@ohsu.edu.



1 and row 2 lower rootlets, and the ANKRD24–TRIOBP complex ensures that rootlets are protected from damage from aging or noise.

## Results

### ANKRD24 surrounds TRIOBP at stereocilia insertion points

We detected ANKRD24 in shotgun mass spectrometry experiments that characterized proteins of isolated chicken and mouse vestibular stereocilia (Shin et al., 2013; Krey et al., 2015). When we examined proteins of FACS-sorted cells from *Pou4f3-Gfp* mice (Krey et al., 2018b), which express GFP exclusively in hair cells (Scheffer et al., 2015), we found that ANKRD24 and TRIOBP were robustly detected in early postnatal GFP-positive cochlea and utricle cells using data-independent acquisition (DIA; Fig. 1, A and B).

Exploiting image-scanning fluorescence microscopy with an Airyscan detector (Müller and Enderlein, 2010; Sheppard et al., 2013; Roth et al., 2016), with a point-spread function (PSF) of 200 nm (Krey et al., 2020), we found that ANKRD24 was prominently associated with the stereocilia rootlet in outer hair cells (OHCs), inner hair cells (IHCs), and vestibular (utricle) hair cells (VHCs) on postnatal day 7.5 (P7.5; Fig. 1, C and D). On P1.5, ANKRD24 was initially present on the apical surface of OHCs and IHCs as well as at the rootlets; the apical labeling eventually disappeared, and most ANKRD24 localized to stereocilia rootlets (Fig. 1, E and G; and Fig. S1, C, F, G, and I). ANKRD24 colocalized at lower rootlets with TRIOBP in OHCs and IHCs (Fig. 1, F–G), and its intensity in rootlets was similar along the tonotopic axis of the cochlea (Fig. S1 A). We generated higher-resolution profile views of the rootlets by folding cochlear tissues and imaging the rootlet axis in a single x–y plane; in P19.5 IHCs (Fig. 1 H) and OHCs (Fig. 1 I), ANKRD24 was concentrated at stereocilia insertion points (Fig. 1, H and I, arrows), while TRIOBP was found throughout the lower rootlet (Fig. 1, H and I, arrowheads).

We also used lattice structured illumination microscopy (lattice SIM; Betzig, 2005; Richter et al., 2019), which in our hands offered a PSF of 150 nm for 488-nm illumination and 176 nm for 568-nm illumination (Fig. S1, P and Q). Lattice SIM allowed visualization of the actin-free channel surrounding the rootlet (Furness et al., 2008; Fig. 1, J–L, arrowheads; Fig. S1 L). In IHCs, the central actin density of the rootlet was surrounded by an unlabeled zone ∼400 nm wide (Fig. 1, O, Q, and S). The full-width at half-maximum (FWHM) of the central actin density was 122 ± 14 nm ($n$ = 6) in experiments with ANKRD24 labeling and 113 ± 20 nm in experiments with TRIOBP ($n$ = 6; P = 0.37 for $t$ test comparison). The concentration of ANKRD24 at the insertion with the signal trailing into the rootlet gave it a comet-like appearance in IHCs and OHCs (Fig. 1, J and K; and Fig. S1, I and J), as well as VHCs (Fig. S1, K and M). While the comet pattern was most prominent past 2 wk, accumulation of ANKRD24 at the insertion point was also seen at earlier ages (Fig. S1, C, F, G, and I).

In single-plane images that captured the hair cell surface, ANKRD24 at the stereocilia insertion point was resolved as a ring (Fig. 1, J and M). Rings at the insertion points were also seen early in development in IHCs and OHCs (Fig. S1, B, D, and H).

Comparing four separate experiments, each averaging >25 ANKRD24 ring images, we found that the ring's peak-to-peak diameter was 183 ± 5 nm (mean ± SD; Fig. 1, N and S). The ANKRD24 ring structure continued through the cuticular plate (Fig. 1 M). ANKRD24 labeling within the cuticular plate was less broad than at the stereocilia insertion, consistent with the ANKRD24 ring tightening around the rootlet deeper in the cuticular plate (Fig. 1, O, P, and S); it was nevertheless considerably wider than the actin core. The peak-to-peak diameter for ANKRD24 in the rootlet region was 136 ± 7 nm ($n$ = 4 sets of >25 images), while the FWHM for a single Gaussian fit was 268 ± 15 nm ($n$ = 6). By contrast, when using the same Alexa Fluor 488 secondary antibody, TRIOBP labeling was more tightly associated with rootlet actin (Fig. 1, Q–S); the FWHM for a single Gaussian fit was 227 ± 28 nm ($n$ = 6; P = 0.0096 for $t$ test comparison to ANKRD24 rootlet profile). These imaging data suggested that TRIOBP is tightly localized around rootlet actin, and ANKRD24 forms an annular shell outside of TRIOBP (Fig. 1 T).

To assess whether ANKRD24 localizes at the membrane in hair cells, we colabeled for ANKRD24, TRIOBP, and the apical membrane linker radixin (RDX) in IHCs and OHCs (Fig. 2, B and C). ANKRD24 at the stereocilia insertion was located exactly at the apical membrane, immediately above cuticular plate actin (Fig. 2, B and C). TRIOBP immunoreactivity extended up the rootlet as far as the ANKRD24 concentration and was relatively uniformly distributed throughout the rootlet's extension through the cuticular plate (Fig. 2, D and E). Exogenous ANKRD24 showed similar targeting; when plasmids encoding ANKRD24 tagged with the HA tag were injectoporated into sensory epithelial explants (Xiong et al., 2014), HA labeling was detected at stereocilia rootlets (Fig. S1, N and O) of hair cells.

### ANKRD24 localizes to plasma membrane and cytoplasmic aggregates in HeLa cells

To examine ANKRD24 interactions, we expressed ANKRD24-GFP in HeLa cells. GFP signal was found at the plasma membrane (marked with wheat germ agglutinin [WGA]; Fig. 2, F and G, arrows) and in large whorl-like aggregates (Fig. 2 F, inset). While N-terminal tagging qualitatively reduced ANKRD24 plasma membrane localization, C-terminally tagged ANKRD24 mimicked untagged ANKRD24 (Fig. S2, A–E). When tagged with GFP or Myc, TRIOBP-5 formed circular aggregates in the HeLa cell cytoplasm when expressed by itself (Fig. 2, H and I; and Fig. S2, F–I). TRIOBP-5 aggregates were usually smaller than ANKRD24 aggregates (Fig. 2, F and H), although TRIOBP-5 aggregates varied in size and appearance (Fig. S2, I–K). Quantitation of ANKRD24-GFP labeling showed that it interacted with WGA-marked plasma membranes; GFP–TRIOBP-5 did not (Fig. 2 J).

Formation of cytoplasmic aggregates suggested that ANKRD24 interacts with itself. We demonstrated with coimmunoprecipitation experiments that ANKRD24-HA interacted with ANKRD24-GFP in HEK293 cells (Fig. 2 K). As a negative control, ANKRD24-HA did not interact with TRIOBP-4-GFP, which includes only the N-terminal half of TRIOBP-5 (see Fig. 3).

WGA, which binds membrane glycolipids and glycoproteins, labeled structures within the ANKRD24 aggregates

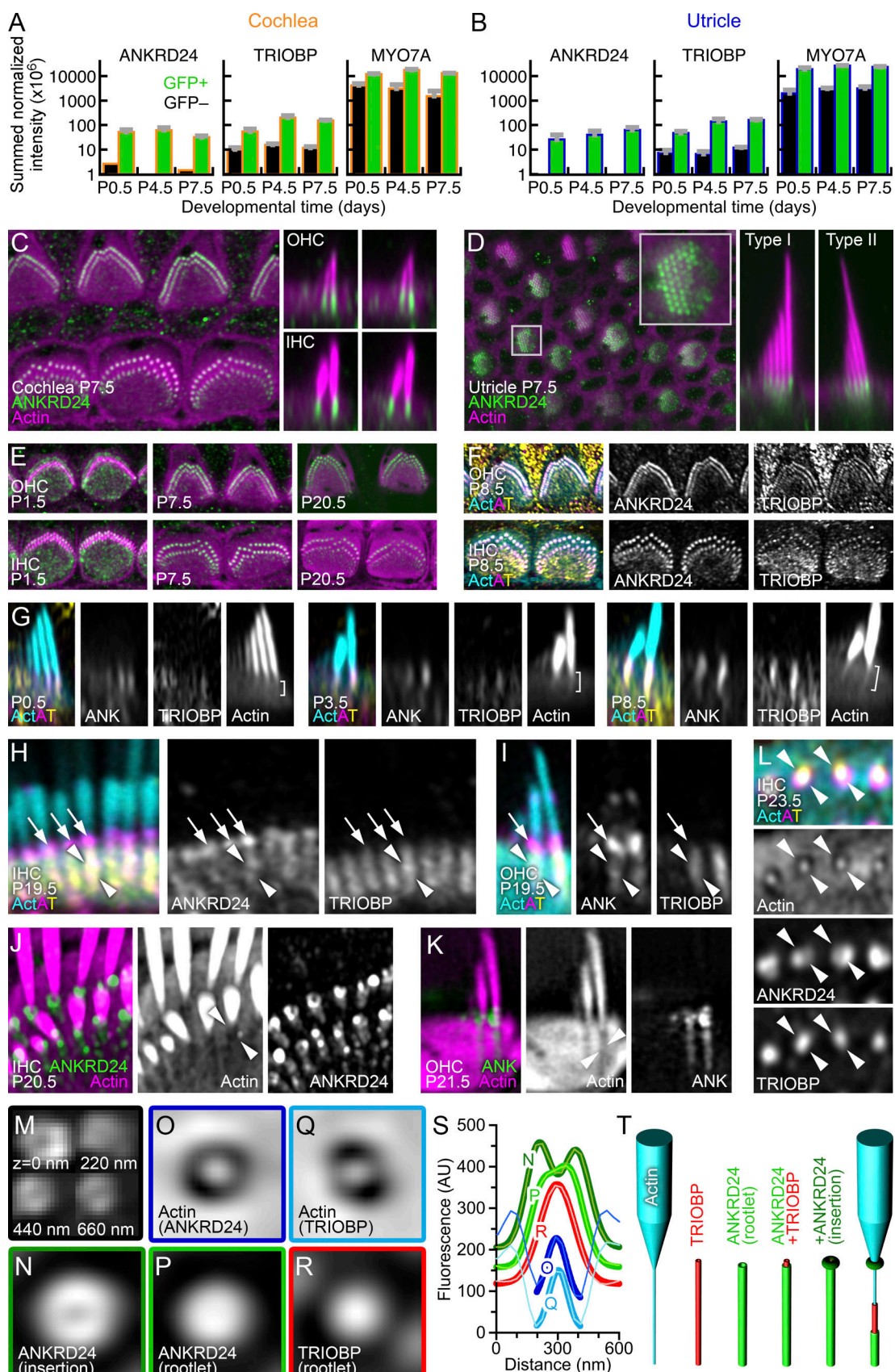

Figure 1. **ANKRD24 localization to hair cell stereocilia rootlets. (A and B)** Mass spectrometry quantitation from cochlea (A) and utricle (B) using DIA. *Pou4f3*-GFP–positive hair cells (green) and –negative cells (black). Mean ± SEM (*n* = 4). **(C)** ANKRD24 staining in midcochlea on P7.5. Left, horizontal view

through rootlets. Right, reslices of OHCs (top) or IHCs (bottom). **(D)** ANKRD24 staining in P7.5 utricle. Left, horizontal view through rootlets; inset, magnified view of the rootlets. Right, reslice images from type I (left) or type II (right) hair cells. **(E)** Developmental progression of ANKRD24 in midcochlea (OHCs, top; IHCs, bottom). Image intensities adjusted to compare labeling. **(F)** Triple labeling of OHCs and IHCs for actin (cyan; Act), ANKRD24 (magenta; A), and TRIOBP (yellow; T). Right panels, ANKRD24 and TRIOBP. **(G)** Development of the rootlet and associated ANKRD24 and TRIOBP labeling from P0.5 through P8.5. Brackets in actin panels highlight rootlet development. **(H and I)** ANKRD24 overlap with TRIOBP in IHCs and OHCs. Arrows indicate position of ANKRD24 at apical surface, and arrowheads indicate ANKRD24 and TRIOBP in the lower rootlet. **(J)** ANKRD24 rings surrounding the membrane insertion in IHCs. Lower-intensity staining at rootlet in actin-free channel (arrowheads). **(K)** ANKRD24 in OHC. ANKRD24 at membrane insertion and extension through rootlet (arrowheads). **(L)** Cross-section of stereocilia rootlets within cuticular plate, near apical surface. ANKRD24 fills the actin-free region, while TRIOBP is more tightly associated with rootlet F-actin. **(M)** Montage of ANKRD24 labeling (Alexa Fluor 488 secondary antibody) from stereocilia insertion (z = 0 nm) in single x–y images at 220-nm intervals through the rootlet. ANKRD24 rings are apparent not only at insertion, but also within the rootlet. **(N–R)** Averages of actin (O and Q) or antibody labeling with Alexa Fluor 488 secondary antibody in row 1 IHC stereocilia (N, P, and R). **(N)** ANKRD24 at stereocilia insertions (average of 40 images). **(O)** Actin from ANKRD24 at rootlets ($n = 50$). **(P)** ANKRD24 at rootlets ($\geq$440 nm below insertions; $n = 50$). **(Q)** Actin from TRIOBP at rootlets ($n = 35$). **(R)** TRIOBP at rootlets ($\geq$440 nm below insertions; $n = 50$). **(S)** Line scans (31.3 nm wide) through center of average images in N–R. Offsets are applied to N and O for clarity. Data (lighter colored points) are fitted with single (actin, TRIOBP) or double (ANKRD24) Gaussians (thick lines). **(T)** Cartoon showing arrangement of actin (truncated stereocilium with rootlet), TRIOBP, and ANKRD24 in rootlet area. Last panel shows all three proteins but with rootlet ANKRD24 and TRIOBP partially removed to show their concentric relationship relative to actin. Panel full widths: C left, 25 µm; C right, 4.5 µm; D left, 44 µm (inset, 5 µm); D right, 4.5 µm; E, 17 µm; F, 18 µm; G and K, 3 µm; H, 4 µm; J, 1.8 µm; L, 2.1 µm; M, 2 × 2 montage of 470-nm panels; N–R, 625 nm. Superresolution modality: A–I, Airyscan; J–R, lattice SIM. AU, arbitrary unit.

---

after permeabilization of the cell (Fig. 2 L). Both WGA labeling and actin were found within the ANKRD24 whorl aggregates (Fig. 2 M, inset). By contrast, while actin was closely associated with TRIOBP-5 aggregates, WGA labeling was not (Fig. 2 N). Thus, ANKRD24 associates with membrane structures within HeLa cells; it also self-associates and colocalizes with TRIOBP-5, the longest TRIOBP splice product, which also interacts with itself (Katsuno et al., 2019).

## The C-terminal domain of ANKRD24 interacts with the C-terminal domain of TRIOBP

Colocalization of ANKRD24 with TRIOBP in hair cells (Figs. 1 and 2) suggested that these two proteins might interact. When expressed in HEK293 cells, either Myc–TRIOBP-1 or Myc–TRIOBP-5 coimmunoprecipitated with ANKRD24-HA (Fig. 3, A and B). Consistent with those results, Myc–TRIOBP-1 or Myc–TRIOBP-5 were colocalized with ANKRD24-HA in large whorls in HeLa cells (Fig. 3, C and E). By contrast, Myc–TRIOBP-4 did not colocalize with ANKRD24-HA (Fig. 3 D). Coexpressed ANKRD24 and TRIOBP-5 were colocalized at filopodia bases in HeLa cells (Fig. 3 F). We validated these observations by quantifying the interaction of TRIOBP splice forms with ANKRD24 using 3D volume rendering with Imaris. The fraction of TRIOBP volumes that had some overlap with ANKRD24 surfaces (overlap fraction, Fig. 3 G) and the ratio of overlapping TRIOBP volume with ANKRD24 volumes (overlap ratio, Fig. 3 H) were comparable for TRIOBP-1 and TRIOBP-5 but reduced for TRIOBP-4. ANKRD24 thus interacts with TRIOBP splice isoforms that contain domains encoded by the 3′ end of the gene.

To test the roles of the different domains of ANKRD24, we generated constructs (Fig. 4 A) that deleted the N-terminal 12 amino acids (ΔN); the ankyrin repeats (ΔANK); the alanine-rich region, which overlaps with the second coiled-coil domain (ΔAla); the C-terminal 331 amino acids (Trunc2); the C-terminal 695 amino acids (Trunc3); the N-terminal 213 amino acids (Trunc4); the N-terminal 213 amino acids and the C-terminal 331 amino acids (Trunc5); or the N-terminal 488 amino acids (Trunc6). In coimmunoprecipitation experiments in HEK293 cells, ANKRD24 Trunc2 and Trunc5, which are missing the

C-terminal half of the second coiled-coil region (residues 845–985), no longer interacted with TRIOBP-1 (Fig. 4 B, highlighted in blue).

When coexpressed in HeLa cells, ANKRD24 deletion constructs were distributed similarly with (Fig. 4, G–O; and Fig. S3, N–V) or without (Fig. 4, D–F; and Fig. S3, A–H) TRIOBP-5 expression. TRIOBP-5 colocalized with ΔN, ΔANK, ΔAla, Trunc4, and Trunc6 constructs, each of which included the C-terminal 140 amino acids of ANKRD24 (Fig. 4, H–J, M, and O; and Fig. S3, O–Q, T, and V). Consistent with the coimmunoprecipitation data, TRIOBP-5 did not colocalize with Trunc2, Trunc3, or Trunc5, each of which lack the Ala and CC2 domains (Fig. 4, K, L, and N; and Fig. S3, R, S, and U). The fraction of TRIOBP-5 volumes that had some overlap with ANKRD24 surfaces (Fig. 4 P) and the overlap ratio of TRIOBP-5 and ANKRD24 (Fig. 4 Q) were both reduced with Trunc2, Trunc3, and Trunc5.

## ANKRD24 domains for self-association and membrane localization

In self-association coimmunoprecipitation experiments in HEK293 cells, Trunc2-HA and Trunc6-HA each interacted with ANKRD24-GFP, and Trunc2-HA and Trunc6-GFP interacted with each other (Fig. 4 C). Plasma membrane localization of ΔN was reduced in HeLa cells, although intracellular aggregates still contained membrane and actin (Fig. 4, D, E, and H; and Fig. S3 I). ΔANK formed very large aggregates within the cytoplasm that included neither membrane nor actin (Fig. 4, F and I; and Fig. S3 J); by contrast, distribution of ΔAla was very similar to that of WT ANKRD24 (Fig. 4 J; and Fig. S3, C and Q). Trunc2 and Trunc3, each lacking the C-terminus but containing the N-terminus and ANK domains, were strongly expressed along the entire membrane (Fig. 4, K and L; and Fig. S3, D, E, R, and S). Trunc5 was largely cytoplasmic, although it did localize to some filopodia; it did not colocalize with TRIOBP-5 (Fig. 4 N; and Fig. S3, G and U). Trunc6 was not located at the plasma membrane (Fig. 4 O; and Fig. S3, H and V) but was quite variable in its appearance (Fig. S3, K–M); in some cells, it was largely cytosolic (Fig. S3 K), but in others, it appeared in large aggregates (Fig. 4 O; and Fig. S3 M), most like ΔANK. Plasma membrane localization of ANKRD24

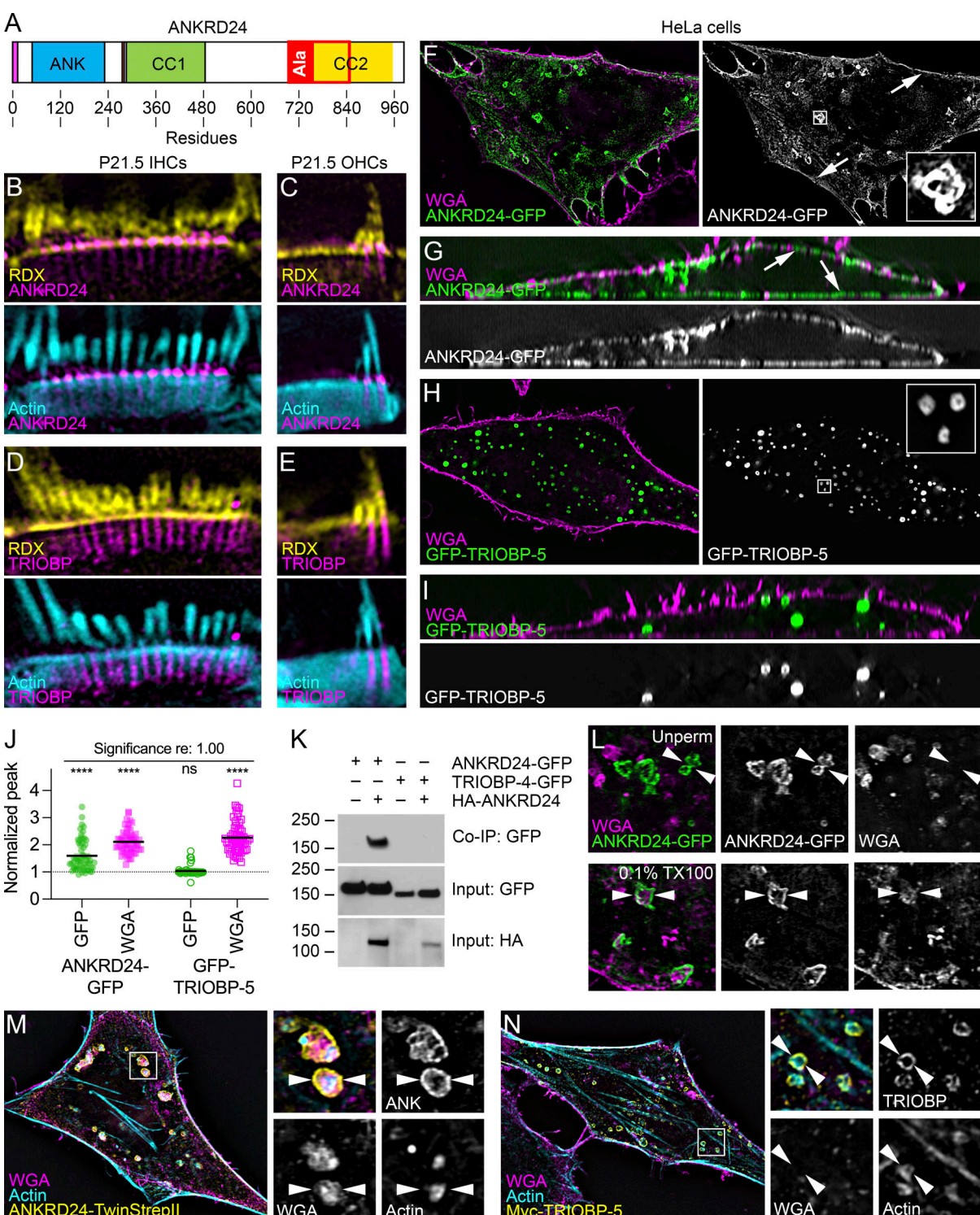

Figure 2. **ANKRD24 self-association and association with plasma membrane and intracellular membranes. (A)** Domain structure of ANKRD24. Magenta box, amino acids 1–12; ANK, ankyrin repeats (five); CC1, coiled-coil region no. 1; CC2, coiled-coil region no. 2; Ala, alanine-rich domain (overlaps with CC2). **(B–E)** Localization of ANKRD24 and TRIOBP relative to RDX and actin (triple-labeled samples) in IHCs and OHCs. **(F–I)**, Localization of ANKRD24-GFP or GFP–TRIOBP-5 in unpermeabilized HeLa cells relative to WGA. Arrows in F and G indicate ANKRD24 at the plasma membrane. **(J)** Quantitation of membrane association (one-sample t tests; n = 60 for each). ****, P < 0.0001. **(K)** Coimmunoprecipitation (Co-IP) of ANKRD24-GFP with HA-ANKRD24. TRIOBP-4-GFP, which does not bind ANKRD24, included as a control. Molecular mass markers (kD) on left. **(L)** Localization of ANKRD24-GFP and WGA in unpermeabilized (left) and permeabilized (right) HeLa cells. WGA labeling is inside ANKRD24 whorl-like aggregates (arrowheads). **(M and N)** Localization of WGA binding and actin relative to ANKRD24-TwinStrepII or Myc–TRIOBP-5. WGA and actin are associated with ANKRD24, but only actin is within TRIOBP-5 aggregates (arrowheads). Panel full widths: B and D, 10 µm; C and E, 5 µm; F and H, 60 µm (insets 3 µm); G and I, 52 µm; L, 10 µm; M and N, 50 µm (panels on right, 6 µm). Superresolution modality for B–I and M–N, lattice SIM. Source data are available for this figure: SourceData F2.

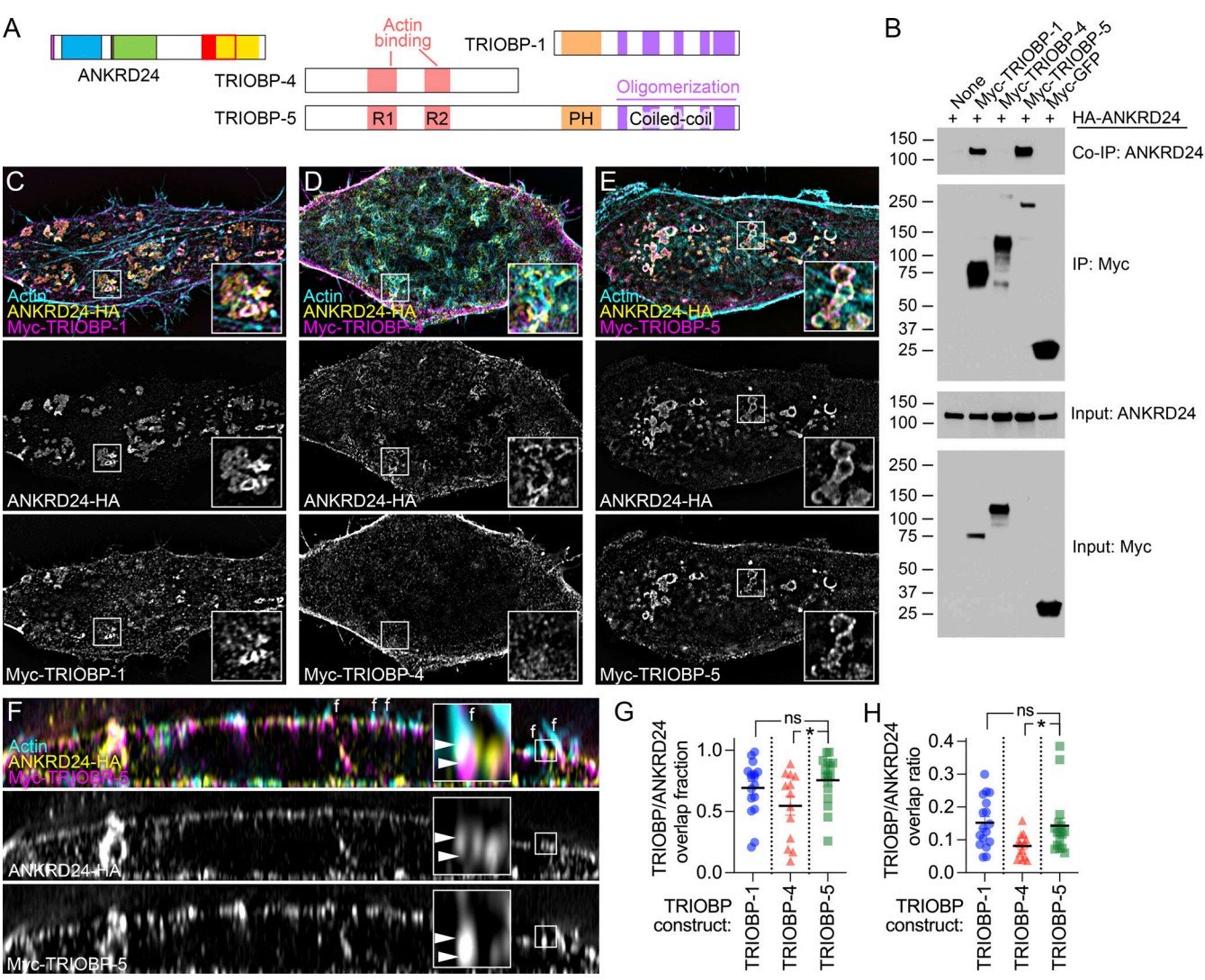

Figure 3. **ANKRD24 interacts with TRIOBP. (A)** ANKRD24 and major TRIOBP splice products. **(B)** ANKRD24-HA coimmunoprecipitates with Myc–TRIOBP-1 and Myc–TRIOBP-5, but not Myc–TRIOBP-4. Immunoprecipitations use anti-Myc. Co-IP shows anti-HA signal, and IP panel shows anti-Myc; input panels show anti-HA and anti-Myc. Markers (in kD) indicated. **(C–E)** Colocalization of ANKRD24-HA with Myc–TRIOBP-1 (C), Myc–TRIOBP-4 (D), and Myc–TRIOBP-1 (E). TRIOBP-1 and TRIOBP-5 colocalized with ANKRD24 in whorl-like aggregates. **(F)** Reslice images showing colocalization of ANKRD24-HA with Myc–TRIOBP-5. ANKRD24 and TRIOBP-5 colocalized at filopodia bases (marked f; see inset). Arrowheads indicate TRIOBP labeling. **(G and H)** Quantitation of TRIOBP splice form overlap fractions (G) and overlap ratios (H) with ANKRD24; t test comparisons to TRIOBP-5; n = 13–17 for each. *, P < 0.05. Panel full widths: C–E, 50 μm (insets, 5 μm); F, 40 μm (inset, 1.5 μm). Superresolution modality for B–F, lattice SIM. Source data are available for this figure: SourceData F3.

thus requires both the N-terminus and the ANK domain; moreover, loss of the ANK domain led to formation of much larger aggregates.

### *Ankrd24^{KO/KO}* mouse exhibits progressive hearing loss

We used CRISPR/Cas9 to engineer a null allele of the *Ankrd24* gene by targeting exons 6 and 17 using separate gRNAs. Several mouse lines with large deletions spanning the two exons were obtained, each of which eliminated a large part of the *Ankrd24* coding sequence (Fig. 5 A). The protein product of the chosen allele is predicted to be truncated after the second ankyrin repeat (Fig. 5 B). *Ankrd24^{KO}* mice were backcrossed to C57BL/6J for more than six generations to eliminate possible off-target modifications; genotyping results from the first backcross are shown in Fig. 5 C.

We confirmed the complete loss of immunolabeling of OHC and IHC rootlets in *Ankrd24^{KO/KO}* mice using either a goat anti-ANKRD24 antibody (Fig. 5, D and E) or four separate rabbit anti-ANKRD24 antibodies (Fig. S4, A–L). One of the rabbit antibodies (no. 213) was targeted to amino acid residues 34–50 (Fig. S4), which are outside the coding region targeted by CRISPR/Cas9 (Fig. 5, A and B); no signal was observed (Fig. S4, E, K, and L), confirming that *Ankrd24^{KO}* is a null allele.

We used auditory brainstem response (ABR) and distortion-product otoacoustic emission (DPOAE) measurements to determine the effect of *Ankrd24* deletion on total auditory function and OHC activity, respectively (Fig. 5, F–H). When measured at ~2 mo, ABR thresholds were elevated at high frequencies in

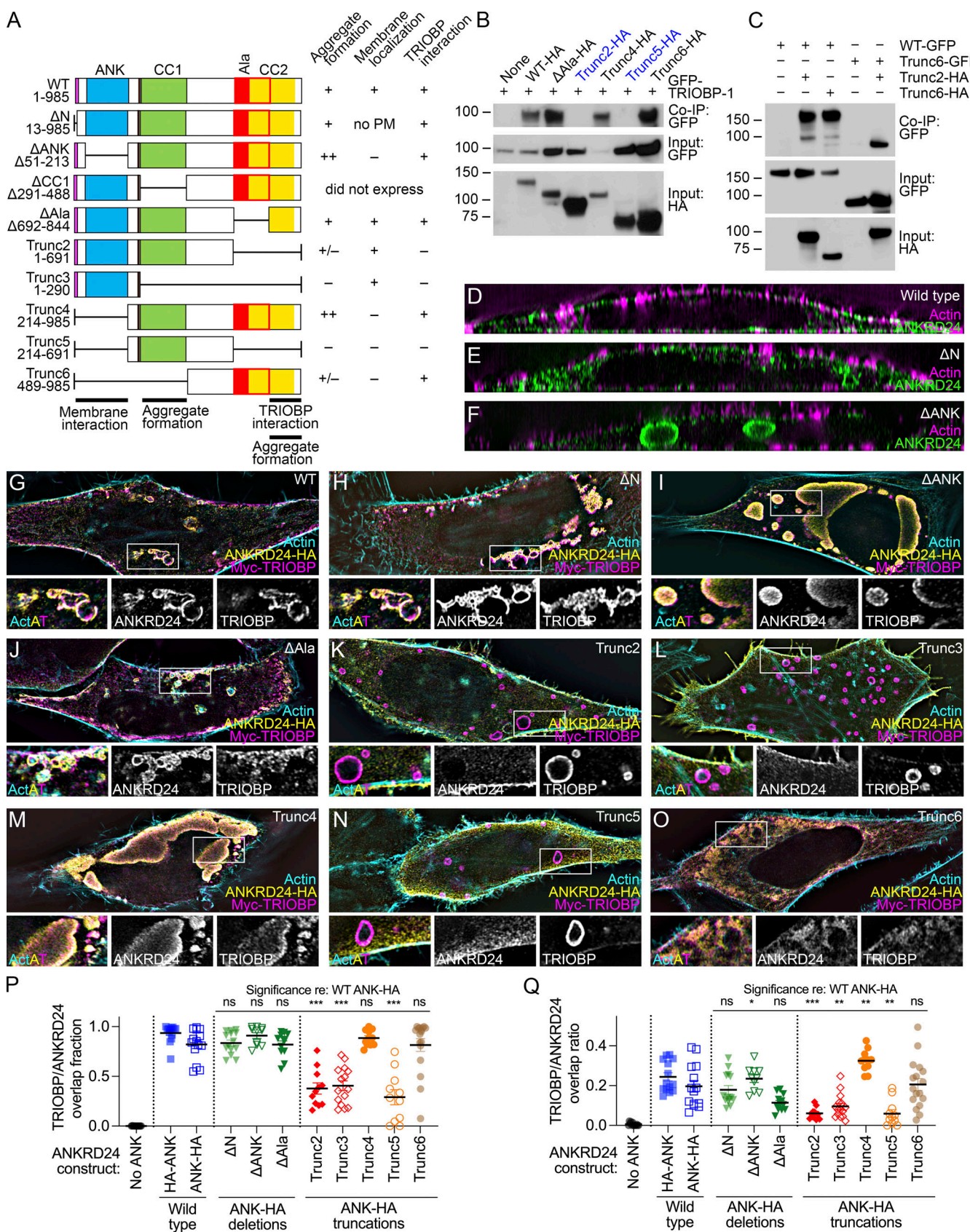

Figure 4. **Mapping functional domains of ANKRD24. (A)** Truncation and deletion constructs. No PM, associated with intracellular membrane but not plasma membrane. For aggregate formation: +, smaller aggregates, similar to WT; ++, very large aggregates; +/−, less clear aggregates. **(B)** Co-IP of GFP–TRIOBP-1 with truncation and deletion constructs. **(C)** Self-association of ANKRD24 truncation constructs. Markers (in kD) indicated. **(D–F)** Reslice images

showing expression of ANKRD24 constructs alone in HeLa cells. **(G–O)** Flat-view images showing coexpression of HA-tagged truncation and deletion ANKRD24 constructs with Myc–TRIOBP-5 in HeLa cells. Magnified view with separated channels below. **(P and Q)** Quantitation of TRIOBP-5 and ANKRD24 overlap fractions (P) and overlap ratios (Q); *t* test comparisons to ANK-HA; *n* = 11–17 for each. *, P < 0.05; **, P < 0.01; ***, P < 0.001. Panel full widths: D–O, 60 μm (insets in G–O, 10 μm). Superresolution modality for D–O, lattice SIM. Source data are available for this figure: SourceData F4.

*Ankrd24*$^{KO/KO}$ mice, with no response detected at 32 kHz (Fig. 5 F). By 6 mo, ABR thresholds were at or near the instrument limit at all frequencies (Fig. 5 G), although hair cells were present and hair bundles were largely intact (Fig. 5, H and I). At 2 mo, DPOAE responses to stimuli of ≥24 kHz were substantially reduced (Fig. 5 J). DPOAE responses of *Ankrd24*$^{KO/+}$ heterozygotes were near those of WT at <40 kHz but showed hearing loss above, suggesting that a single copy of *Ankrd24* was insufficient to maintain high-frequency hearing in OHCs. Vestibular evoked potentials (VsEPs) at 2 mo were not significantly different between the three genotypes (Fig. 5 K).

## Morphological changes in *Ankrd24*$^{KO/KO}$ hair bundles

Stereocilia remained cohesive in *Ankrd24*$^{KO/KO}$ hair bundles, but the bundles themselves were sometimes warped (Fig. 6, A–H). Warped bundles appeared as early as P1.5 for IHCs (Fig. 6 B) but were less obvious in OHCs until P7.5 (Fig. 6 H). IHC bundles were even more disorganized on P7.5 (Fig. 6 F). We identified three classes of bundle warping (wave, mustache, and split) distinct from the WT U-shaped bundle and found that each class appeared more frequently in *Ankrd24*$^{KO/KO}$ hair cells than in heterozygotes (Fig. 6 I). Bundle warping in *Ankrd24*$^{KO/KO}$ bundles increased along the apical-basal cochlear axis on P7.5 and

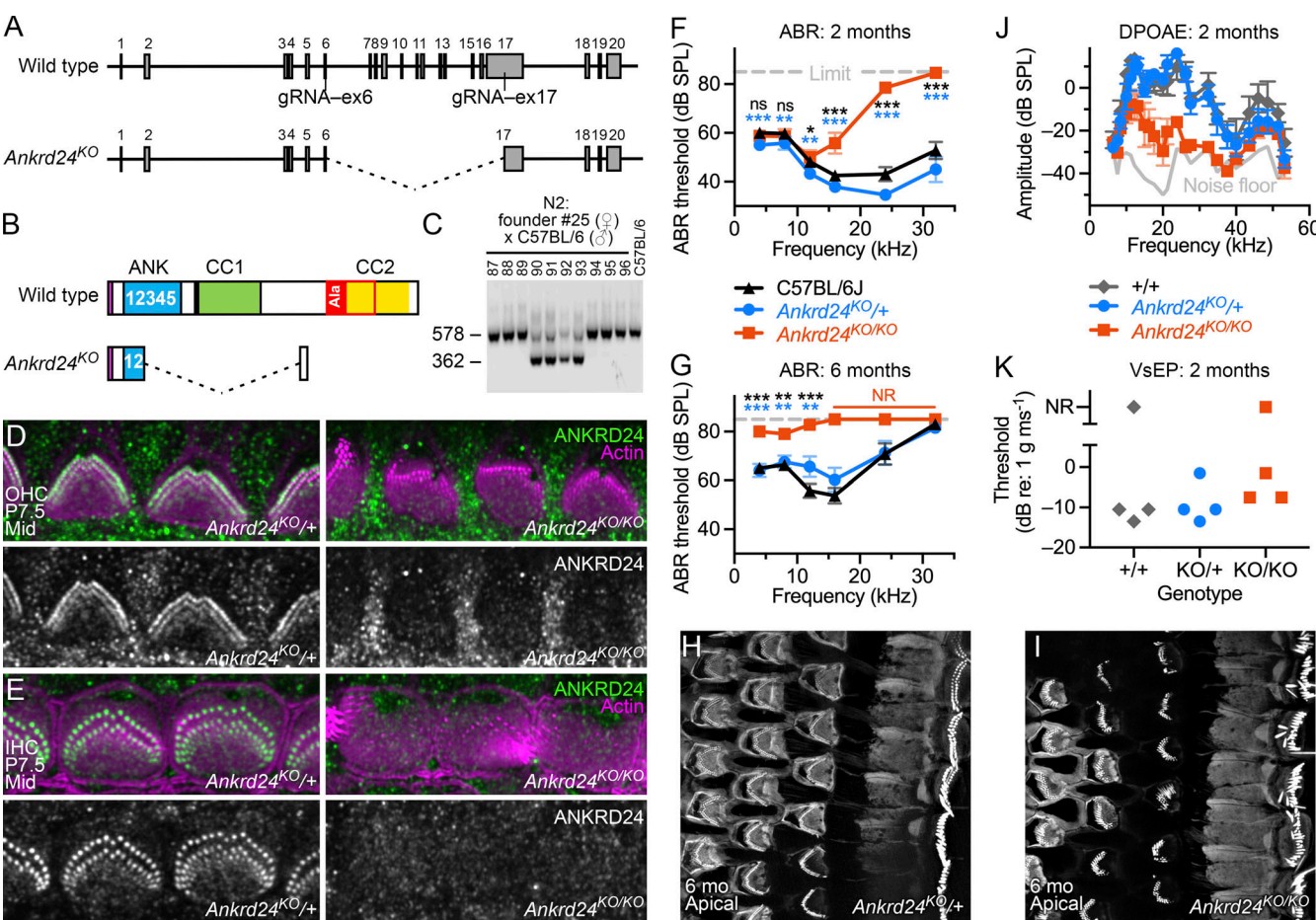

Figure 5. **Characterization of *Ankrd24* mouse null allele. (A)** Gene structure for WT *Ankrd24* (top) and *Ankrd24*$^{KO}$ null allele (bottom). Exons and sites targeted by gRNAs are shown. **(B)** ANKRD24 protein structure. *Ankrd24*$^{KO}$ should produce a truncated protein product. **(C)** Genotyping N2 backcross (female Δex6-17 founder crossed with C57BL/6J male) using primers to detect Δex6-17 deletion. Animals 90–93 have Δex6-17 deletion. **(D)** Immunocytochemistry of OHCs in *Ankrd24*$^{KO/+}$ (left) and *Ankrd24*$^{KO/KO}$ mice (right). **(E)** IHCs from *Ankrd24*$^{KO/+}$ and *Ankrd24*$^{KO/KO}$ mice. **(F and G)** ABR thresholds (mean ± SEM) for *Ankrd24* genotypes at the age of 2 mo (F) and 6 mo (G). Symbols indicating P values are black for *Ankrd24*$^{KO/+}$-C57BL/6J comparison and blue for *Ankrd24*$^{KO/KO}$-*Ankrd24*$^{KO/+}$; *t* tests with *n* = 12–20. *Ankrd24*$^{KO/KO}$ were clearly different from controls but completely lacked responses, preventing statistical testing. *, P < 0.05; **, P < 0.01; ***, P < 0.001. **(H and I)** Phalloidin staining of apical cochlea at 6 mo for *Ankrd24*$^{KO/+}$ (H) and *Ankrd24*$^{KO/KO}$ (I) mice. **(J)** DPOAE amplitudes for *Ankrd24* genotypes; mean ± SEM (*n* = 4). DPOAE amplitudes for *Ankrd24*$^{KO/KO}$ mice were significantly different from *Ankrd24*$^{KO/+}$ by *t* test (P < 0.01) at 17.6, 20.0, 23.8, 26.2, and 32.4 kHz. **(K)** VsEP thresholds for *Ankrd24* genotypes (no significant differences for any genotype pairs; *n* = 4). Panel full widths: D and E, 24 μm; J and K, 50 μm. Superresolution modality: D and E, Airyscan; J and K, SIM.

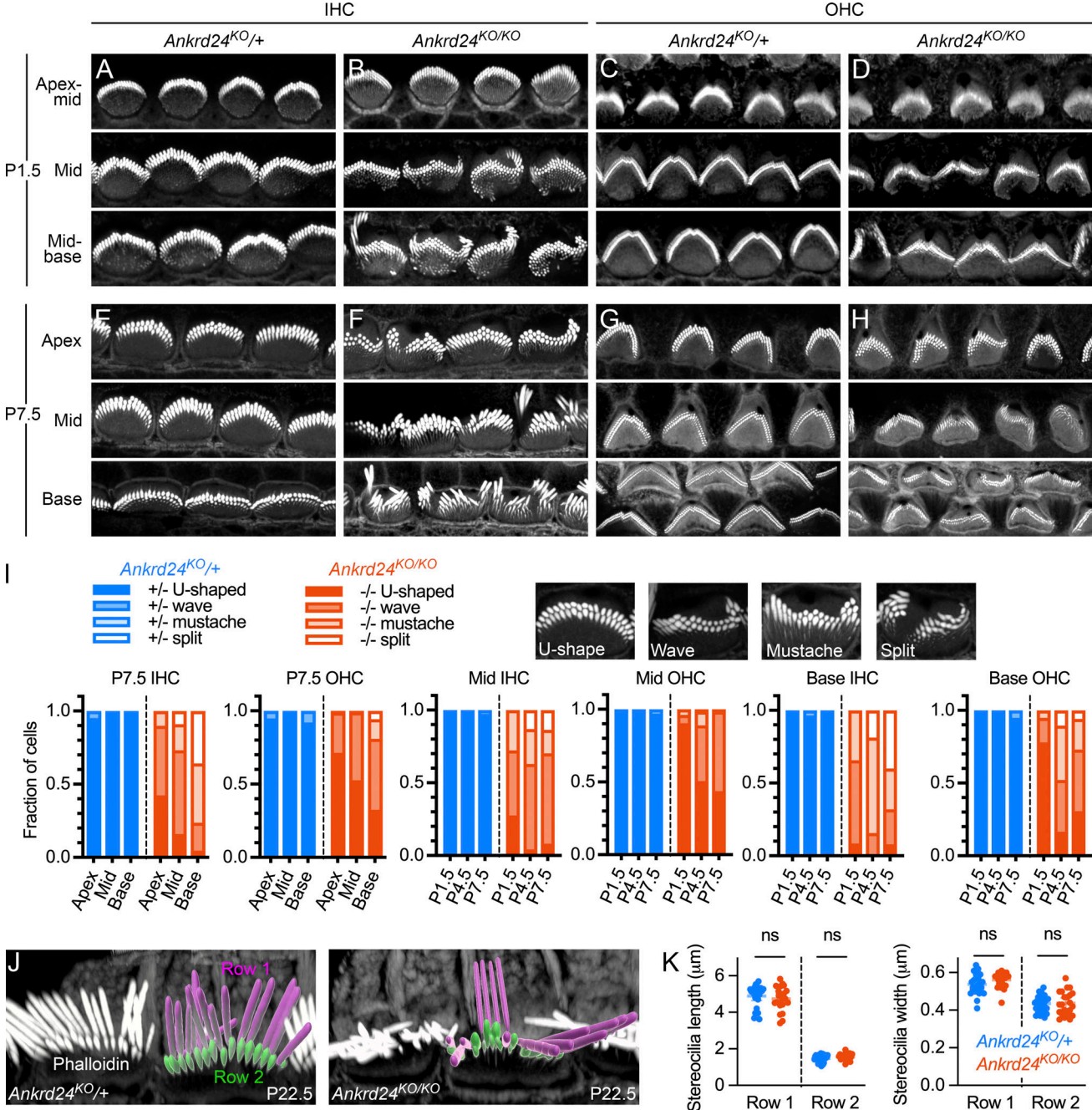

Figure 6. **Hair bundles and stereocilia in *Ankrd24^{KO/KO}* cochleas. (A–H)** Phalloidin labeling for F-actin on P1.5 (A–D) and P7.5 (E–H); IHCs (A, B, E, and F) and OHCs (C, D, G, and H) for *Ankrd24^{KO/+}* and *Ankrd24^{KO/KO}*. **(I)** Bundle warping. Insets, four phenotypes of bundles in control and mutant hair cells; bar graphs, developmental and spatial progression of the phenotype, with *n* = 48–65. **(J)** Surface rendering in Imaris for quantitation of stereocilia dimensions. **(K)** Stereocilia dimension quantitation. Using *t* tests (*n* = 24–26), no significant differences in row 1 or 2 length or width were found between *Ankrd24^{KO/+}* control and *Ankrd24^{KO/KO}* mutant stereocilia. Panel full widths: A and B, 3 μm; C and E, 7.5 μm; G, 10 μm (insets, 1 μm); H, 8 μm; J, 20 μm; L and M, 3 μm. Superresolution modality for A–J, Airyscan.

developmentally across the first postnatal week in basal IHCs and OHCs (Fig. 6, A–I). We measured stereocilia dimensions using surface rendering and quantitation (Fig. 6, J and K) and found no significant changes in length or width for either row 1 or row 2 stereocilia (Fig. 6 K).

Transmission EM (TEM) on P6.5 indicated that the osmiophilic rootlet was slightly wider in *Ankrd24^{KO/KO}* cochleas than in normal-hearing *Ankrd24^{KO/+}* controls (Fig. 7, A–D); in addition, hollow rootlets were more frequent in *Ankrd24^{KO/KO}* hair cells than in controls (arrow, Fig. 7 D). Wider rootlets and hollow rootlets were more obvious on P21.5 (Fig. 7, E–H). In WT and *Ankrd24^{KO/+}* controls, the actin-free channel surrounding the lower rootlet was apparent (asterisks in Fig. 7, E and F; and Fig. S5 U); filaments were often seen extending from the rootlet to

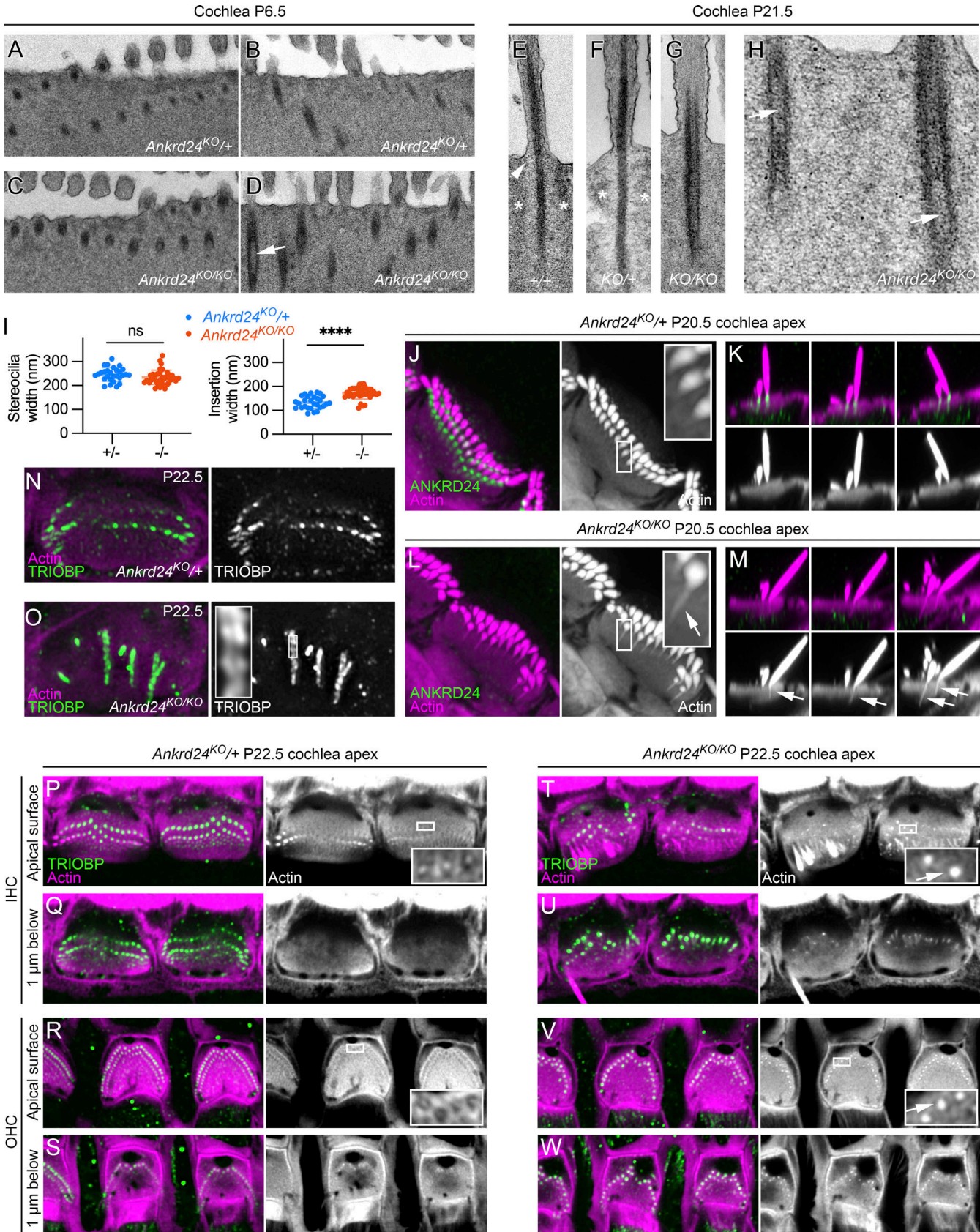

Figure 7.   **Rootlet F-actin expands in *Ankrd24^{KO/KO}* hair cells. (A–D)** TEM of *Ankrd24^{KO/+}* (A and B) or *Ankrd24^{KO/KO}* (C and D) cochleas on P6.5. Arrow indicates hollow rootlet. **(E–H)** P21.5 +/+ control (E), *Ankrd24^{KO/+}* (F), and *Ankrd24^{KO/KO}* (G and H) rootlets. Arrowhead in E, filaments between rootlet and

plasma membrane. Asterisks in E and F, extent of low-density region surrounding rootlet (absent in G and H). Arrows in H, hollow rootlets. **(I)** Insertion width (membrane-to-membrane distance at narrowest point) and stereocilia width (membrane-to-membrane distance above taper) in P21.5 electron micrographs; *t* test comparisons with *n* = 26–36. ****, P < 0.0001. **(J–M)** Thickened rootlets in *Ankrd24*$^{KO/KO}$ hair cells (arrows in L and M). **(N and O)** Horizontal slices through cuticular plate showing abnormal rootlets in *Ankrd24*$^{KO/KO}$ hair cells (O). **(P–W)** Cuticular plates and rootlets from *Ankrd24*$^{KO/+}$ (P–S) or *Ankrd24*$^{KO/KO}$ (T–W) cochleae at or below the apical surface. Arrows in insets indicated thickened rootlets in *Ankrd24*$^{KO/KO}$ hair cells. Panel full widths: A–D, 2.5 µm; E–G, 0.75 µm; H, 0.82 µm; J and L, 10 µm (insets, 1 µm); K and M, 8 µm; N and O, 10 µm (inset 1 µm); P–W, 20 µm (insets, 1.7 µm). Superresolution modality: J–M and P–W, Airyscan; N and O, lattice SIM.

the plasma membrane (arrowhead, Fig. 7 E). In *Ankrd24*$^{KO/KO}$ hair cells, however, lower rootlets were thicker, and there was no space between rootlets and the cuticular plate (Fig. 7, G and H; and Fig. S5 V). Hollow rootlets were very common (Fig. 7 G and H) and often extended into the upper rootlet within stereocilia (Fig. 7 G). Measurement of distances using P21.5 TEM images showed that whereas the stereocilia shaft diameter was unchanged in *Ankrd24*$^{KO/KO}$ hair cells, the width of the stereocilia insertion was significantly greater than that of insertions of *Ankrd24*$^{KO/+}$ controls (Fig. 7 I).

In comparison to *Ankrd24*$^{KO/+}$ control hair cells (Fig. 7, J, K, and P–S), *Ankrd24*$^{KO/KO}$ hair cells labeled with phalloidin had much more prominent actin in the lower rootlet but no actin-free channel between rootlet and cuticular plate actin; rootlet actin appeared to have expanded to fill this region (Fig. 7, L, M, and T–W). These results corroborated the TEM images showing wider lower rootlets.

Thickened rootlets were most often apparent in row 1 stereocilia of *Ankrd24*$^{KO/KO}$ mice (Fig. 7, M and T–W). Rootlets often no longer ran vertically in the cuticular plate; some instead curved within or below the cuticular plate (Fig. 7, N–O; and Fig. 8, D and F). Widened rootlets could also be seen with TRIOBP labeling (Fig. 7 O, inset). Altered TRIOBP localization could be seen in IHCs and OHCs across development (Fig. S5, A–F, and O) and from apex to base on P7.5 (Fig. S5, I–N). TRIOBP was still present in rootlets of VHCs, but as in cochlea, it was distributed nonuniformly (Fig. S5, Q–T). We also noted increased gaps in actin labeling between the cuticular plate and the cell borders in *Ankrd24*$^{KO/KO}$ hair cells, especially OHCs (Fig. 7, V and W).

### ANKRD24 is essential for normal distribution of TRIOBP-5

The intracellular location of TRIOBP shifted in *Ankrd24*$^{KO/KO}$ hair cells. In controls, TRIOBP labeling extended to the apical plasma membrane as marked by RDX labeling (Fig. 8 A). By contrast, TRIOBP labeling was weaker and no longer immediately adjacent to the plasma membrane in *Ankrd24*$^{KO/KO}$ hair cells (Fig. 8 B). TRIOBP labeling intensity was also more variable in *Ankrd24*$^{KO/KO}$ hair cells than in controls (Fig. 8, C–F). We quantified TRIOBP signal in hair cells by surface rendering the immunofluorescence signal, then analyzing surface dimensions with Imaris software (Fig. 8, G–O). The average fraction of stereocilia in a cell with substantial TRIOBP labeling was significantly smaller in mutants than in controls, especially in row 2 (Fig. 8 I). While the average length, width, and intensity of the TRIOBP signal changed relatively modestly in *Ankrd24*$^{KO/KO}$ hair cells (Fig. 8, J–L), the variability in those measurements (coefficient of variation [CV]) was higher in the mutants (Fig. 8, M and O).

### TRIOBP-5 recruits ANKRD24 to stereocilia rootlets

We also examined ANKRD24 localization in *Triobp* mutant lines. Because a *Triobp-1* knockout is embryonic lethal (Kitajiri et al., 2010), we used two previously generated mouse lines that lack TRIOBP-5 (Katsuno et al., 2019). In the *Triobp*$^{Δex9-ex10}$ line, where exons specific for TRIOBP-5 are deleted without consequence for TRIOBP-1 or TRIOBP-4 expression, ANKRD24 labeling was absent from rootlets of homozygous-null IHCs and OHCs (Fig. 8, P–S) and VHCs (Fig. S5, AA and AB). The warped hair-bundle phenotype was present in these hair cells, albeit less prominent than in *Ankrd24*$^{KO/KO}$ mice (Fig. S5, A–C). We also examined *Triobp*$^{Δex8/YHB226}$ compound heterozygous mice; one allele each of WT TRIOBP-1 and TRIOBP-4 are expressed in these mice, but each allele of TRIOBP-5 is truncated before the C-terminal coiled-coil domains (Katsuno et al., 2019). ANKRD24 labeling was also missing from rootlets of *Triobp*$^{Δex8/YHB226}$ compound heterozygous mice (Fig. S5, W–Z). TRIOBP-5 is thus necessary for ANKRD24 localization at the stereocilia insertion point, and the coiled-coil domains of TRIOBP likely contain an interaction site for ANKRD24.

To determine directly whether TRIOBP-5 localizes ANKRD24 in rootlets, we used a Helios gene gun to transfect DsRed–TRIOBP-5 into hair cells of P2-P3 homozygous *Triobp*$^{Δex9-ex10}$ mouse organ of Corti explants, which lack TRIOBP-5. In transfected hair cells, DsRed–TRIOBP-5 not only incorporated into lower rootlets of stereocilia but also recruited endogenous ANKRD24 to lower rootlets (Fig. 8, T–W, arrows). By contrast, stereocilia of neighboring untransfected TRIOBP-5 deficient hair cells lacked ANKRD24 at their bases (Fig. 8, T, V, and W, arrowheads). Thus, mislocalization of ANKRD24 in the *Triobp*$^{Δex9-ex10}$ mouse was reversed by introduction of DsRed-tagged TRIOBP-5, which restored ANKRD24 to its normal location in rootlets.

### Vulnerability of *Ankrd24*$^{KO/KO}$ stereocilia

To assess whether ANKRD24 protects hair cells from noise damage, mice were exposed to octave band limited noise at 8–16 kHz (96 dB) for 2 h; in C57BL/6 mice, this treatment results in a moderate temporary loss of auditory function, which recovered to near baseline within 2 wk. ABRs recorded from *Ankrd24*$^{KO/KO}$ mice recovered to a lesser degree after noise exposure than heterozygous and C57BL/6J control mice (Fig. 9, A–F). Measurement of the difference between ABR thresholds before (baseline measurements) and 2 wk after noise exposure gave an estimate of the permanent shift in threshold (Fig. 9, G–I). Both control genotypes showed substantial but largely nonsignificant shifts at high frequencies (Fig. 9, G and H); *Ankrd24*$^{KO/KO}$ mice already had profound hearing loss at these frequencies and thus showed no shift (Fig. 9 I). By contrast, *Ankrd24*$^{KO/KO}$ mice showed a significant threshold shift at 8–16 kHz (Fig. 9 I); the control genotypes showed no permanent shift there (Fig. 9, G and H).

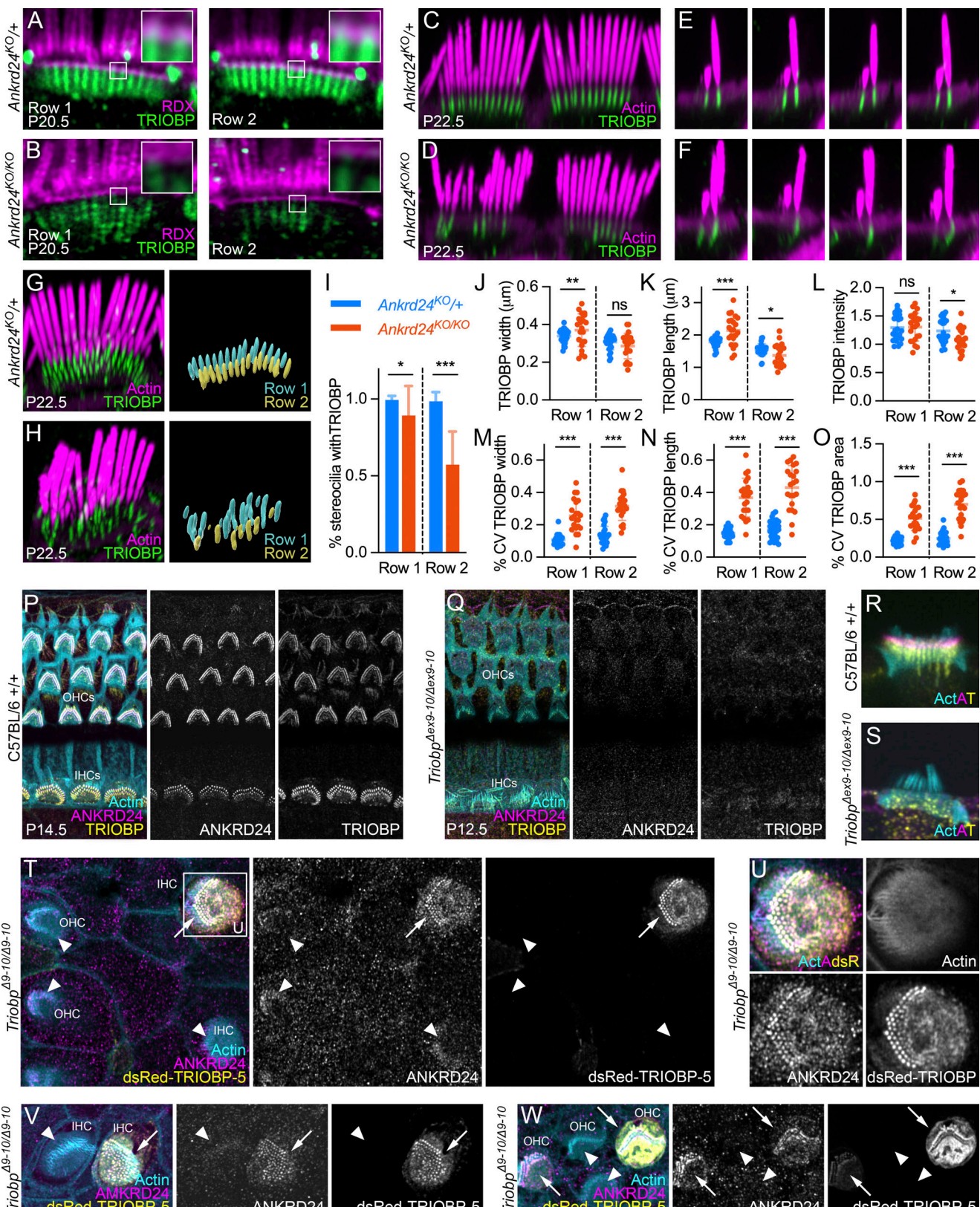

Figure 8. **Interdependence of ANKRD24 and TRIOBP. (A and B)** Profile view of TRIOBP in relation to RDX. Insets, TRIOBP is close to apical surface membrane in control but separated in mutants. **(C and D)** En face view of TRIOBP immunoreactivity in *Ankrd24*[KO/+] (C) and *Ankrd24*[KO/KO] (D) IHCs. **(E and F)** Reslice view of TRIOBP immunoreactivity in *Ankrd24*[KO/+] (E) and *Ankrd24*[KO/KO] (F) IHCs. **(G and H)** Left, immunofluorescence labeling. Right, Imaris surface rendering of TRIOBP immunolabeling. **(I–O)** Quantitation of TRIOBP signal in rows 1 and 2; *t* test comparisons with *n* = 22–28. **(I)** Fraction of stereocilia with detectable TRIOBP. *, P < 0.05; ***, P < 0.001. **(J–O)** TRIOBP measurements and variability. Each point is average data from one cell (excluding stereocilia with

undetectable TRIOBP). *, P < 0.05; **, P < 0.01; ***, P < 0.001. **(J)** Width of TRIOBP staining. **(K)** Length of TRIOBP staining. **(L)** Intensity of TRIOBP staining. **(M)** CV for width measurements. **(N)** CV for length measurements. **(O)** CV for area measurements. **(P–S)** ANKRD24 is present in rootlets of WT cochlear hair cells (P and R) and absent from $Triobp^{\Delta ex9-ex10/ex\Delta 9-ex10}$ ($Triobp^{\Delta 9-10/\Delta 9-10}$) cochlear hair cells (Q and S). **(T–W)** Hair cells of $Triobp^{\Delta 9-10/\Delta 9-10}$ with introduced dsRed–TRIOBP-5 recruit ANKRD24 to rootlets (arrows); untransfected cells do not (arrowheads). Low-power view (T), magnified transfected IHC from T (U), transfected and untransfected IHCs (V), and OHCs (W). Panel full widths: A and B, 10 µm (insets, 1 µm); C and D, 20 µm; E and F, 6 µm; G and H, 10 µm; P and Q, 30 µm; R and S, 8 µm; T, 35 µm; U, 30 µm; V, 10 µm; V and W, 21.5 µm. Superresolution modality for A–H and P–W, Airyscan.

Noise damage did not dramatically affect bundles in $Ankrd24^{KO/+}$ or $Ankrd24^{KO/KO}$ mice (Fig. 9, J–M).

Although no obvious defects in stereocilia structure were seen by scanning EM on P21.5 (Fig. 9, N and O), the total number of row 1 stereocilia decreased in $Ankrd24^{KO/KO}$, suggesting that these stereocilia may be vulnerable to damage (Fig. 9 P). In addition, stereocilia were often missing from $Ankrd24^{KO/KO}$ cochleas that had been prepared for immunocytochemistry, and far more stereocilia were isolated from $Ankrd24^{KO/KO}$ cochleas by glass adsorption than from those of heterozygote controls. Stereocilia thus fractured more readily in $Ankrd24^{KO/KO}$ animals.

When hair bundles were stimulated with a fluid jet, $Ankrd24^{KO/KO}$ IHCs had mechanoelectrical transduction (MET) currents that were like those of heterozygous controls (Fig. 10, A, B, and J), albeit with consistently steeper activation curves (Fig. 10, C and K). Bundles were subjected to a 12-s overstimulation protocol with an amplitude 120% of that required to produce a maximum transduction current (Fig. 10 E). In control cells, current amplitudes diminished by ~25%; by contrast, amplitudes from $Ankrd24^{KO/KO}$ diminished by >80% (Fig. 10, D and M). At the end of the overstimulation period, while $Ankrd24^{KO/+}$ hair cells had normal MET responses, $Ankrd24^{KO/KO}$ hair cells responded at double the stimulus frequency (Fig. 10 E), which may reflect uncoordinated bundle motion after damage. In P10–P11 animals, row 1 stereocilia of $Ankrd24^{KO/KO}$ bundles were often disarrayed after overstimulation; in P12 mice, the protocol often completely eliminated row 1 and 2 stereocilia (Fig. 10, F–I). $Ankrd24^{KO/KO}$ stereocilia are thus more sensitive to mechanical damage.

## Discussion

ANKRD24 is a key component of the stereocilia rootlet and is dependent on the presence of TRIOBP-5 for its localization to the

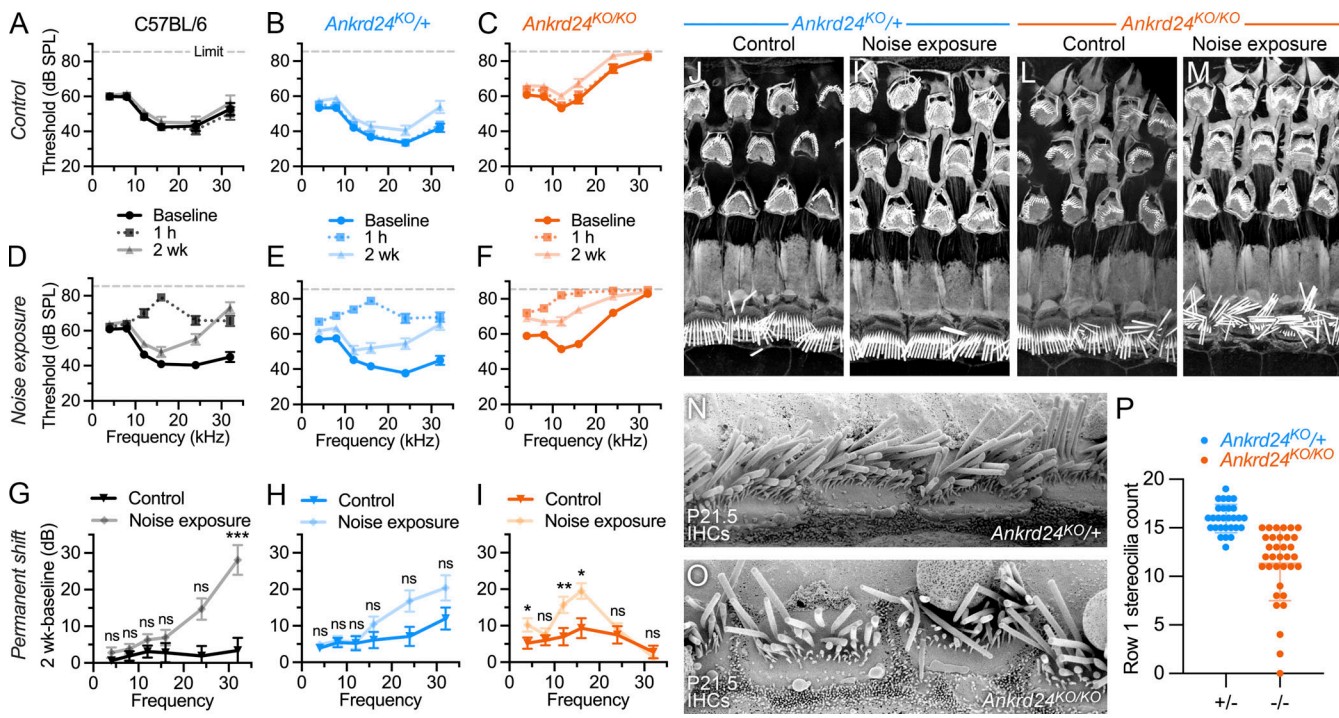

**Figure 9. Noise damage impacts $Ankrd24^{KO/KO}$ hair cells. (A–C)** Control conditions. Mice were tested for baseline ABR function (solid dark lines), then were subjected to ambient sound. Temporary threshold shift was measured 2 h after treatment (dotted line); permanent threshold shift was measured 2 wk after treatment (solid light lines). Gray dashed line, system limit. **(D–F)** Noise exposure. Mice treated identically to control mice except that treatment was octave band-limited noise (96 dB SPL at 8–16 kHz). **(G–I)** Permanent shift: difference in threshold between ABRs at 2 wk and baseline for both control and noise treatments. Cox proportional hazard model used to determine significance of permanent shift; n = 16–30. *, P < 0.05; **, P < 0.01; ***, P < 0.001. **(J–M)** Actin in apical cochlea for control (J and L) and noise exposure (K and M) treatments. **(N and O)** Scanning EM of IHCs in $Ankrd24^{KO/+}$ and $Ankrd24^{KO/KO}$ cochleas on P21.5. **(P)** Stereocilia count in row 1 from scanning EM images of $Ankrd24^{KO/+}$ and $Ankrd24^{KO/KO}$ cochleas on P21.5; t test comparison (n = 34). Panel full widths: H–N, 30 µm. Superresolution modality for H–L, lattice SIM.

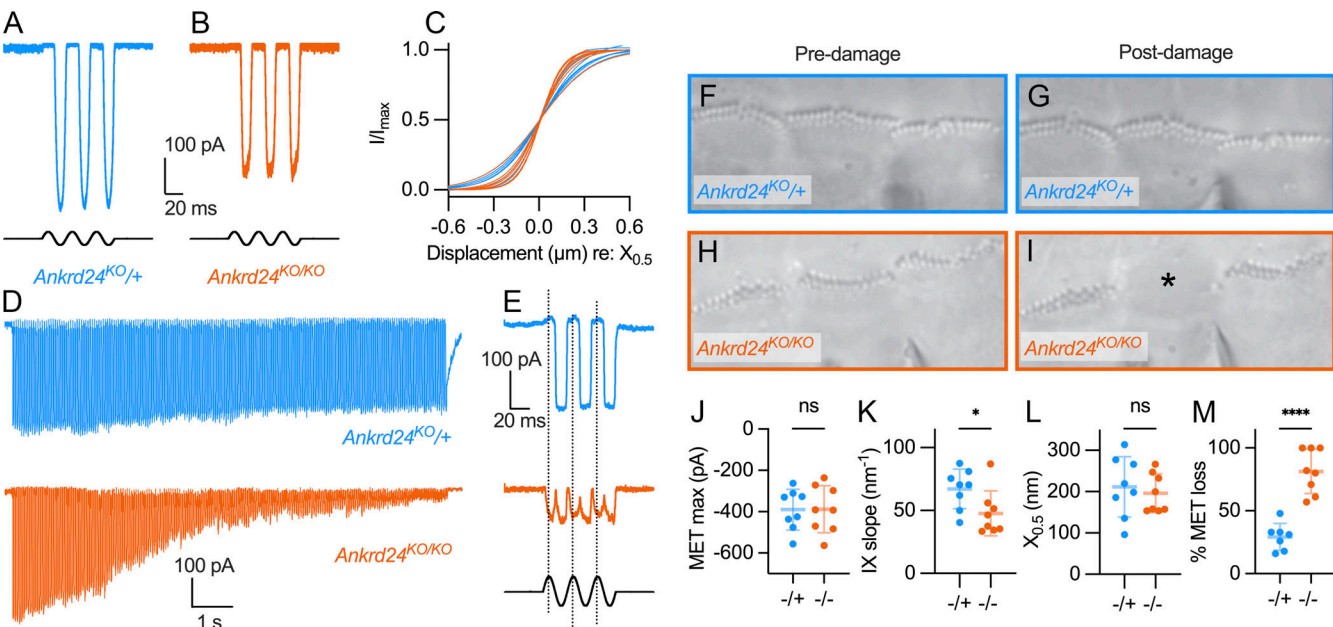

Figure 10. ***Ankrd24^{KO/KO}* hair bundles are sensitive to overstimulation.** Sensitivity of MET and stereocilia to hair-bundle overstimulation in P10–P12 mouse IHCs. **(A and B)** Examples of MET from *Ankrd24^{KO/+}* (A) and *Ankrd24^{KO/KO}* (B) IHCs. **(C)** Overlaid Boltzmann fits to IX curves from eight *Ankrd24^{KO/+}* and eight *Ankrd24^{KO/KO}* IHCs. Curves normalized by dividing current (I) by maximum current ($I_{max}$); aligned with $X_{0.5} = 0$ to show differences in slope. **(D)** Example currents in response to overstimulation (167 Hz; 120% amplitude). Scale bars apply to both traces. **(E)** Example traces taken near end of prolonged stimulus; double-frequency (2f) response of *Ankrd24^{KO/KO}*. **(F–I)** Differential interference contrast micrographs of IHC bundles taken before (F and H) and after (G and I) prolonged stimuli. Asterisk in I, loss of stereocilia. Panel full widths are 38 μm. **(J–M)** Summary plots; *t* test comparisons; *n* = 7–8. **(J)** MET amplitude. **(K)** Activation (IX) curve slope (nm for e-fold slope change). *, P < 0.05. **(L)** Activation curve midpoints. **(M)** MET loss during prolonged stimulation. ****, P < 0.0001.

lower rootlet but necessary for normal TRIOBP-5 distribution among and along rootlets. In chicken cochlea, rootlet growth occurs during Tilney's stage 3 (Tilney et al., 1992), which occurs from P0 to P6 in mouse hair cells (Krey et al., 2020). Mouse rootlets grow then (Kitajiri et al., 2010), associated with the appearance of ANKRD24 and TRIOBP-5 along the rootlet (Fig. 1). Loss of ANKRD24 leads to fragile stereocilia; accordingly, *Ankrd24^{KO/KO}* mice have high-frequency hearing loss that progresses to near-complete deafness by 6 mo of age. The progressive hearing loss is similar to but slower than hearing loss in mice lacking TRIOBP-5, suggesting that the vulnerability of the rootlets when ANKRD24 is absent results from abnormal maintenance of TRIOBP-5.

## Domain structure of ANKRD24

ANKRD24 belongs to the N-Ank family of proteins, which are hypothesized to shape membranes by producing positive curvature (Wolf et al., 2019). Stereocilia insertion sites have high positive membrane curvature, and ANKRD24's concentration there later in development may allow for stabilization of such an energetically unfavorable structure. Alternatively, ANKRD24 may be targeted to the taper membrane by an unusual lipid composition (Zhao et al., 2012). As with RAI14 (Wolf et al., 2019), the N-terminal 12 amino acids increase targeting of ANKRD24 to the plasma membrane. The ANK repeats contribute to membrane interaction, too, and constructs including these domains sequester WGA-labeled structures, which are likely membranous, in aggregates of various sizes (Fig. 2). In the absence of the

ANK repeats, ANKRD24 aggregates grow very large and sequester neither membrane nor actin (Figs. 4 and S3). ANKRD24 contains two regions of predicted coiled-coil structure, CC1 and CC2, each about 200 amino acids long. Recent predictions of human (Uniprot Q8TF21) and mouse (Uniprot Q80VM7) ANKRD24 structure using the AlphaFold 2.0 algorithm (Jumper et al., 2021) suggested that CC2 interacts intramolecularly with the ANK domain, which could allow more control of membrane interaction and self-association.

Deletion constructs allowed us to identify key domains of ANKRD24 (Fig. 4). Immunostaining experiments in heterologous cell lines showed that ANKRD24 formed large aggregates that logically require interactions between more than one pair of self-association domains (Fig. 2). Large aggregates formed reliably only with constructs containing both CC1 and at least the C-terminal half of CC2 (Fig. 4, G–J and M; and Fig. S3, A–M). Coimmunoprecipitation experiments showed that Trunc2 and Trunc6 each interact with full-length ANKRD24 (Fig. 4 C). In addition, Trunc2-HA, which contains CC1 but lacks CC2, interacts with Trunc6-GFP, which contains CC2 but not CC1 (Fig. 4 C). Trunc6-HA expressed by itself formed large aggregates but only in some cells, apparently those with higher expression levels (Fig. S3, K–M). This result suggests that CC2 can interact with itself, perhaps more weakly than the presumed CC1–CC2 interaction. TRIOBP also interacted with CC2; the ΔAla ANKRD24 construct, which lacks the first part of CC2, still interacts with TRIOBP (Fig. 4, B and J), suggesting that the interaction requires only the C-terminal 141 amino acids. Whether self-association of

ANKRD24 through CC2 prevents TRIOBP-5 binding and vice versa is not known.

## ANKRD24 and TRIOBP-5 cooperate to assemble the rootlet

The rootlet has a remarkable coaxial arrangement of its proteins. Actin filaments of the rootlet are packed together unusually tightly (Itoh and Nakashima, 1980; Itoh, 1982), wrapped by TRIOBP (Kitajiri et al., 2010). We show here that ANKRD24 surrounds TRIOBP, as the overall apparent diameter of ANKRD24 in the rootlet (~270 nm) is significantly greater than that of TRIOBP (~230 nm; Fig. 1, M–S). Measurement across the ANKRD24 ring allows us to estimate its actual diameter, which is ~140 nm; although the precision of this measurement is influenced by the PSF, its accuracy is not. There is yet another coaxial structure: after P10, the spectrin subunits SPTAN1 and SPTBN1 also form a ring around the rootlet (Liu et al., 2019). Superresolution microscopy showed that the spectrin ring diameter was 200–210 nm, suggesting it encircles the ANKRD24 ring. The rootlet thus appears to contain coaxial actin, TRIOBP, ANKRD24, and spectrin.

ANKRD24 relies on TRIOBP-5 for rootlet targeting, while reciprocally, ANKRD24 controls the rootlet distribution of TRIOBP-5. Although *Triobp*$^{\Delta ex9-\Delta ex10}$ mice lacked ANKRD24 within rootlets, the normal localization of ANKRD24 could be restored by introduction of exogenous *DsRed–Triobp-5* cDNA into mutant hair cells (Fig. 8). In the absence of ANKRD24, TRIOBP-5 organization within rootlets is disrupted; the close connection of TRIOBP-5 to the apical surface is lost (Fig. 8, A and B), the fraction of stereocilia with TRIOBP-5 in their rootlets is diminished (Fig. 8, G–I), and features of the TRIOBP-5 signal in the rootlet become far more variable (Fig. 8, M–O). The organization of actin within the rootlets of *Ankrd24*$^{KO/KO}$ stereocilia was also disrupted. In both TEM and superresolution images, rootlets were often thicker and hollow, especially in row 1 stereocilia (Fig. 7). These results suggest that the annular shell of ANKRD24 surrounding TRIOBP-5 along the rootlet may restrict and stabilize the size of the rootlet actin bundle; ANKRD24 may also alter the function of TRIOBP-5 to allow for tighter actin bundling. Similarly, when ANKRD24 was coexpressed with TRIOBP-5 in HeLa cells, TRIOBP-5 aggregates became smaller and more condensed compared with the circular and often hollow aggregates seen with TRIOBP-5 alone (Figs. 2 and 3).

Together, these results suggest that ANKRD24 and TRIOBP-5 coalesce around actin filaments that are destined to form the foundation of a mature and durable rootlet. Although ANKRD24 is recruited to the rootlet by TRIOBP-5, the presence of ANKRD24 enables the proper maturation of the rootlet. ANKRD24 maintains the normal uniform distribution of TRIOBP-5 along and among the rootlets, especially those in rows 1 and 2. In addition, ANKRD24's ability to interact with membranes suggests that ANKRD24 accumulates there in part to link TRIOBP-5 to the plasma membrane.

## Physiological function of ANKRD24

In response to loud noise, stereocilia rootlets tend to fracture at the insertion point (Tilney et al., 1982; Liberman, 1987), suggesting fragility there. To ensure that stereocilia can pivot at their bases but be protected against easy fracturing, the cell may permit rootlet actin filaments to slide with respect to each other (Howard and Ashmore, 1986; Kitajiri et al., 2010; Pacentine et al., 2020). In accordance with this model, TRIOBP wraps rootlet actin filaments unusually closely together (Kitajiri et al., 2010), which may account for the lack of tight actin crosslinkers in the rootlet (Krey et al., 2016b). In the absence of ANKRD24, many rootlets are greater in diameter and apparently contain more actin, no longer forming a compact bundle. Consequentially, *Ankrd24*$^{KO/KO}$ stereocilia could become susceptible to fracture or be more easily extracted.

By anchoring the rootlet radially to the cell membrane, ANKRD24 creates a broad surface area of interaction that may mechanically stabilize stereocilia. ANKRD24 appears to reinforce the rootlet at the narrow insertion point, and its ANK repeats could make this structure elastic (Lee et al., 2006); elasticity could allow rootlet reinforcement while still allowing for motion of stereocilia at the base. Without this structure reinforcing the rootlet at the insertion point, the resistance of motion for stereocilia at their base would be reduced; stereocilia of *Ankrd24*$^{KO/KO}$ mice therefore are more susceptible to noise damage or direct overstimulation.

Formation of a tight TRIOBP collar, anchored to the apical membrane by ANKRD24, apparently permits the stereocilia-rootlet combination to successfully respond to countless flexions over the life of a hair cell. Spectrin must play a role in anchoring the rootlet complex to the rest of the cuticular plate; *Sptbn1*$^{KO/KO}$ mice have extremely disorganized rootlets and loss of hearing (Liu et al., 2019). The TRIOBP collar forms with the recruitment and assistance of ANKRD24 early in hair cell development, then later engages spectrin. Without ANKRD24, early postnatal hair bundles become warped, indicating the importance of ANKRD24 for anchoring each stereocilium early. Bundles of hair cells lacking TRIOBP are less misshaped than those in *Ankrd24*$^{KO/KO}$ mice, suggesting that ANKRD24 has a role beyond simply connecting TRIOBP-5 to the membrane. Securing stereocilia rootlets within the cuticular plate thus involves the coordinated actions of multiple proteins.

# Materials and methods

## Mass spectrometry

DIA mass spectrometry data were obtained from a dataset that is described in detail elsewhere (Krey et al., 2018b) and located at https://www.ebi.ac.uk/pride/archive/projects/PXD006240. For each of three runs at each developmental time point, the relative intensities (peptide intensity divided by sum of all identified peptide intensities) of each of the peptides mapped to the indicated protein were summed; the mean ± SEM was plotted ($n = 3$ for each).

## Antibodies

The goat anti-ANKRD24 antibody (E-17) was obtained from Santa Cruz Biotechnology (sc-241811, RRID:AB_10851440), but has since been discontinued. Using Genemed Synthesis, we produced a polyclonal antiserum by injecting four peptides shared by chicken and mouse ANKRD24 into rabbits:

C+QNSVSSHEKQGAPKKR (no. P211; mouse residues 261–276), EQHKERRRKEPLEAEAS+C (no. P212; 315–331), RVAS-LIAHKGLVPTKLD+C (no. P213; 34–50), and C+LYRTHL-LYAIQGQMDEDVQ (no. P214; 950–968). C+ or +C indicates a cysteine residue not found in the native sequence. Peptides were linked to SulfoLink resin (20401; Thermo Fisher Scientific), and antibodies specific for each were separately purified from the antiserum raised against the peptide mixture. The TRIOBP antibody (anti-TARA; 16124-1-AP; RRID:AB_2209237; Proteintech) was raised against the C-terminal 374 amino acids of TRIOBP-5, which are shared with TRIOBP-1. This antibody detects no lower rootlet protein in $Triobp-5^{\Delta ex9–\Delta ex10}$ knockout hair cells, indicating that the lower rootlet signal resulted from TRIOBP-5. The TRIOBP-5 antibody was described previously (Kitajiri et al., 2010); it was raised against a sequence unique to TRIOBP-5. In WT hair cells, we observed a staining pattern with this TRIOBP-5–specific antibody (Fig. S1, I and J) that was identical to that with the anti-TARA antibody, confirming that the latter antibody detects TRIOBP-5 at the lower rootlet (Katsuno et al., 2019). The mouse anti-RDX antibody used for membrane labeling was obtained from Abnova (H00005962-M06, RRID:AB_464027). For HeLa cell immunocytochemistry, we used a monoclonal antibody from IBA (2-1507-001, RRID:AB_513133) to detect the TwinStrepII tag, 6E2 anti-HA antibody (2367, RRID:AB_10691311; Cell Signaling Technology) to detect the HA tag, and a rabbit polyclonal antibody (2272, RRID:AB_10692100; Cell Signaling Technology) to detect the Myc tag. In immunoblotting and immunoprecipitation experiments, antibodies against HA (11867423001, RRID:AB_390918; Roche), Myc (sc-40, RRID:AB_627268; Santa Cruz Biotechnology), and GFP (sc-9996, RRID:AB_627695; Santa Cruz Biotechnology) were used.

## Mice

The *Ankrd24* locus was targeted for CRISPR-mediated knockout (Sander and Joung, 2014) using gRNAs designed to exons 6 and 17 (http://crispr.mit.edu/). The gRNA sequences (exon 6, 5′-GCGTGGTGGACATCGAGGAC-3′; exon 17, 5′-TCAGGCCGAGGTCGTCCCCC-3′) were individually cloned into the DR274 gRNA expression vector (42250; Addgene) and transcribed using the MegaScript T7 kit (Thermo Fisher Scientific). The in vitro–transcribed gRNAs were purified using the NucleoSpin miRNA kit (Macherey-Nagel) and quantified using a NanoDrop spectrophotometer. A mixture containing 30 ng/µl of each gRNA and 100 ng/µl of the Cas9 mRNA (Trilink) was prepared and injected into zygotes of C57BL/6NJ mice (005304; The Jackson Laboratory), and zygotes were transferred into oviducts of pseudopregnant CD-1 females (022; Charles River). Founders were screened for mutations in the targeted exons as well as for large deletions in the intervening sequence.

Founder no. 25 has a large genomic deletion of 6,214 bp. The indel starts with an imperfect insertion of 7 bp (T/GCTTCC) where the initial T, part of the intronic sequence, is the first base variation from the WT genomic sequence. The remaining 6 bp of the insert match the first bases of exon 6, afterward, then a deletion of 6,207 bp removes exons 7–16 and the first 65 bp of exon 17. This deletion results in a frameshift mutation. In the cDNA, the first 6 bp of exon 6, position 327 from the start codon,

joins directly to the last 1,312 bp of exon 17 (cDNA GCTTCC|CCTGGT, where "|" indicates the exon–intron boundary). In translation, after WT Ser109, the amino acids vary from WT for four amino acids (PGLV) until codon 114, which is an early stop that ends translation. This founder line was backcrossed on to the C57BL/6J background for more than six generations and propagated for the experiments described in this study. *Ankrd24* CRISPR mice were genotyped using the following primers: common forward primer, 5′-CTGTGGGTCTCATTTGCCTCAA-3′; WT reverse primer, 5′-GAGAGAGACCCTGTCTCAGAGG-3′; and knockout reverse primer, 5′-GTCTCTTCTCCTTCTTGAGCCC-3′. Note that although the WT reverse primer annealing site is removed in the KO mouse, the knockout primer sequence is in the WT mouse; nevertheless, the 6,586 bp between them does not amplify with Taq polymerase.

The line we used in subsequent experiments (from founder no. 25) was designated as $Ankrd24^{em1Pgg}$ by the Mouse Genome Informatics Database; we refer to it here as $Ankrd24^{KO}$. The $Triobp^{\Delta ex9–ex10}$ mouse line and the $Triobp^{\Delta ex8/YHB226}$ compound heterozygote cross and their genotyping were previously described (Katsuno et al., 2019).

All mouse studies were performed in accordance with the guidelines established by the Institutional Animal Care and Use Committees of Oregon Health & Science University (OHSU; protocol no. IP00000714), the National Institutes of Health (NIH; protocol no. 1263), the University of Nebraska-Lincoln, and Stanford University. Animals were housed in individually ventilated cages (Thoren) with a maximum of five animals per cage with food and water ad libitum. Breeder pairs were separated 16–18 d after crossing; singly housed animals were provided with nesting material; and pregnant dams were given breeder chow (5058 PicoLab; LabDiet) in place of standard chow (5LOD Irr Rodent; LabDiet). Mice were housed under barrier specific pathogen–free conditions in accordance with the OHSU Institutional Animal Care and Use Committee.

## Plasmid constructs

Mouse *Ankrd24* encodes the full-length ANKRD24 protein (Q80VM7; UniProtKB) of 985 amino acid residues (106.2 kD). All reference to specific amino acids of ANKRD24 refer to Q80VM7. *Ankrd24* deletions and HA-tagged Trunc constructs originated from the same *Ankrd24-TwinStrepII* plasmid (VectorBuilder). The original *Ankrd24-TwinStrepII* deletion constructs (ΔN, ΔANK, and ΔAla) were made using an NEB Q5 Site-Directed Mutagenesis Kit (E0554S). For cell culture experiments, the TwinStrepII tag was replaced with an HA tag. The Takara Bio In-Fusion kit (638910) was used for all subsequent cloning; this kit uses PCR primers with 15-bp homology overhangs within a plasmid for deletion or to clone in the sequence of interest. The short HA tag allows its sequence to be incorporated into the primer overhangs; a single inverse PCR therefore removed the TwinStrepII tag and replaced it with an HA tag. Next, the production of the six Trunc-HA constructs was started by making a linear acceptor vector with a C-terminal HA tag with primers to remove ANK repeat sequence. Then an additional six primer sets, with 15-bp homologous ends to the linear acceptor vector, were used to amplify the desired sequence from the original *Ankrd24-*

*TwinStrepII* plasmid. The codons for the amino acids are as follows: Trunc1–AA 1–742, Trunc2–AA 1–691, Trunc3–AA 1–290, Trunc4–AA 214–985, Trunc5–AA 214–691, and Trunc6–AA 489–985. Additional clones were made including a full-length *Ankrd24-HA* plasmid and *Ankrd24 Trunc6-Gfp* plasmid. The full-length *Ankrd24-HA* plasmid was made similarly to the ΔN, ΔANK, and ΔAla constructs, replacing the TwinStrepII with an HA tag. Trun6-GFP was made by deleting codons for amino acids 1–213 in an *Ankrd24-Gfp* plasmid (VectorBuilder).

## Immunocytochemistry

Dissected cochleas or utricles were fixed in 4% formaldehyde (1570; Electron Microscopy Sciences) in dissection buffer (HBSS; 14025126; Thermo Fisher Scientific) with 5 mM Hepes (pH 7.4) for 20–60 min at room temperature. Organs were washed twice in PBS, then cochleas were dissected from periotic bones and the lateral wall was removed. Organs were permeabilized in 0.2% Triton X-100 in 1× PBS for 10 min and blocked in 5% normal donkey serum (017-000-121; Jackson ImmunoResearch) diluted in 1× PBS (blocking buffer) for 1 h at room temperature. Organs were incubated overnight at 4°C with primary antibodies in blocking buffer and then washed three times in 1× PBS. Dilutions were 1:250 for all primary antibodies except for the anti-ANKRD24 goat polyclonal antibody (1:200) and the anti-RDX antibody (1:500). Tissue was then incubated with secondary antibodies: 2 µg/ml donkey anti-goat Alexa Fluor 488 (A-11055, RRID:AB_2534102), 2 µg/ml donkey anti-mouse Alexa Fluor 488 (A-21202, RRID:AB_141607), 2 µg/ml donkey anti-rabbit Alexa Fluor 488 (A-21206, RRID:AB_2535792), 2 µg/ml donkey anti-goat Alexa Fluor 568 (A-11057, RRID:AB_142581), 2 µg/ml donkey anti-mouse Alexa Fluor 568 (A10037, RRID:AB_2534013), or 2 µg/ml donkey anti-rabbit Alexa Fluor 568 (A10042, RRID: AB_2534017; all Thermo Fisher Scientific). CF405 phalloidin (1 U/ml; 00034; Biotium) was also included for the 3–4-h room temperature treatment with secondary antibodies. When only one primary antibody was used, organs were incubated as above with Alexa Fluor 488 secondary antibody and 0.4 U/ml CF568 phalloidin (00044; Biotium). Tissue was washed three times in PBS and mounted on a glass slide in ~50 µl Vectashield (H-1000; Vector Laboratories) and covered with a no. 1.5 thickness 22 × 22-mm cover glass (2850-22; Corning).

HeLa cells (CCL-2, RRID:CVCL_0030; American Type Culture Collection [ATCC]) were maintained in Eagle's minimal essential medium (30-2003; ATCC) supplemented with 10% serum (Serum Plus II; 14009C; Sigma-Aldrich) and 10 ml/l penicillin-streptomycin (P4333; Sigma-Aldrich) in a humidified 5% (vol/vol) $CO_2$ incubator at 37°C. The HeLa cell line was authenticated by ATCC and was also found to be free of mycoplasma contamination while in use for experiments (mycoplasma detection kit 30-1012K; ATCC). Cells were grown on acid-washed no. 1.5 thickness 22 × 22–mm cover glasses (2850-22; Corning) placed in 6-well plates and coated with 0.01 mg/ml poly-L-lysine hydrobromide (P1274; Sigma-Aldrich). Cells were transfected at ~60–70% confluency with Lipofectamine 3000 (L3000008; Invitrogen) following the manufacturer's protocol and using 3.75 µl lipofectamine and 2.5 µg total plasmid DNA per well. 24–36 h after transfection, cells were fixed in 4% formaldehyde (1570;

Electron Microscopy Sciences) in PBS for 15 min at room temperature and rinsed twice in PBS before staining. Cells were permeabilized in 0.1% Triton X-100 in 1× PBS for 10 min and blocked in 2% BSA (A3803; Sigma-Aldrich) diluted in 1× PBS (blocking buffer) for 30 min to 1 h. Cells were incubated overnight at 4°C or for 1.5–3 h at room temperature with primary antibodies diluted 1:250 in blocking buffer, then washed three times in 1× PBS. Cells were then incubated with secondary antibodies: 2 µg/ml donkey anti-mouse Alexa Fluor 488 (A-21202, RRID: AB_141607), 2 µg/ml donkey anti-rabbit Alexa Fluor 488 (A-21206, RRID:AB_2535792), 2 µg/ml donkey anti-mouse Alexa Fluor 568 (A10037, RRID:AB_2534013), or 2 µg/ml donkey anti-rabbit Alexa Fluor 568 (A10042, RRID:AB_2534017; all Thermo Fisher Scientific). Alexa Fluor 647 phalloidin (2 U/ml; A22287; Invitrogen) was also included for the 1–2-h room temperature treatment with secondary antibodies. When only one primary antibody was used, cells were incubated as above with Alexa Fluor 488 secondary antibody (or left with native GFP fluorescence) and 0.4 U/ml CF568 phalloidin (00044; Biotium). Coverslips were washed three times in PBS and mounted on glass slides using Everbrite mounting media (23005; Thermo Fisher Scientific). For membrane labeling, Hela cells were incubated in 5 µg/ml CF568 WGA (29077-1; Biotium) diluted in 1× PBS for 20 min at room temperature after fixation. Cells were rinsed twice with 1× PBS before proceeding with the above staining procedures. In some cases, cells were incubated with WGA solution after permeabilization.

## Gene-gun transfection experiments

P2–P3 *Triobp*$^{\Delta ex9–\Delta ex10}$ mouse organ of Corti explants were kept in culture overnight, and then transfected via Helios gene gun (Belyantseva, 2016) using 1-µm gold particles coated with *DsRed-Triobp-5* cDNA (Katsuno et al., 2019). They were fixed 24 h later for 40 min at room temperature in 4% formaldehyde (50-980-487; Electron Microscopy Sciences) in 1× PBS and processed for immunostaining using goat or rabbit ANKRD24 antibodies and phalloidin-Atto 390 (50556-10NMOL; Sigma-Aldrich) as described above.

## Injectoporation transfection experiments

Injectoporation was performed as described (Xiong et al., 2014; Liu et al., 2018). In brief, the organ of Corti was isolated and cultured in DMEM/F12 medium (11330057; Life Technologies) for 2–4 h. Glass electrodes (2-µm diameter) were used to deliver the plasmid (500 ng/µl in 1× HBSS) to the sensory epithelium. A series of three pulses were applied at 1-s intervals with a magnitude of 60 V and duration of 15 ms (BTX ECM 830 square wave electroporator). Cochlear explants were cultured in a 37°C incubator for 2–3 d. Samples were then fixed in 4% formaldehyde for 20 min at room temperature and processed for immunostaining using 6E2 anti-HA antibody (2367, RRID:AB_10691311; Cell Signaling Technology). Whole-mount preparations were imaged on a Deltavision Deconvolution Microscope and processed with SoftWoRx software (Applied Precision).

## Fluorescence microscopy

Most Airyscan images were acquired using a 63×, 1.4-NA Plan-Apochromat objective on a Zeiss 32-channel LSM880 laser-

scanning confocal microscope equipped with an Airyscan detector and run under Zen (v2.6, 64-bit; Zeiss) acquisition software. A small number of Airyscan images (Fig. 1, F and G; and Fig. 8, A and B) were acquired using a 63×, 1.4-NA Plan-Apochromat objective on a Zeiss 3-channel LSM980 laser-scanning confocal microscope equipped with an Airyscan.2 detector and run under Zen (v3.1, 64-bit; Zeiss) acquisition software. Settings for x–y pixel resolution, z-spacing, pinhole diameter, and grid selection were set according to software-suggested settings for optimal Nyquist-based resolution. Raw data processing for Airyscan-acquired images was performed using manufacturer-implemented automated settings. Display adjustments in brightness and contrast, reslices, and average Z-projections were made in Fiji/ImageJ software. For cochlea imaging, for each antibody, two to four images were acquired from one to two cochlea per genotype per age for each experiment, and experiments were repeated at least twice. Ears from control and mutant littermates or from different ages of C57BL/6J mice of both sexes were stained and imaged on the same days for each experiment to limit variability. Genotyping was performed either before dissection or performed on tails collected during dissection for younger animals (<P8). Genotypes were known by the experimenter during staining and image acquisition. During image acquisition, the gain and laser settings for the antibody and phalloidin signals were adjusted to reveal the staining pattern in control samples, and the corresponding KO samples used the same settings. Image acquisition parameters and display adjustments were kept constant across ages and genotypes for every antibody/fluorophore combination.

SIM images were acquired with a 63×, 1.4-NA, oil-immersion lens on a Zeiss lattice-based Elyra 7 microscope with dual PCO.edge 4.2 sCMOS cameras for detection. Grid selection and z-spacing were guided by the software and kept consistent across images. Grid spacing was relaxed when imaging CF405 phalloidin, as the illumination pattern lacked modulation and was kept consistent across all images. Post-acquisition processing was performed with software-recommended standard filtering for the 488-, 568-, and 647-nm channels and weak filtering for the 405-nm channel. Processing was performed without baseline subtraction and with "scale to raw" checked. Contrast was manually adjusted to retain both dim and bright structures in channels with high dynamic range. Experimental PSF calculations were performed using the Zen software, and images of 100 nm TetraSpeck beads (T7279; Invitrogen) were collected using the same acquisition and processing settings used for sample imaging. Channel alignment was checked based on imaging of 100-nm bead slides.

For Airyscan imaging of the inner ear tissues from of *Triobp^{Δex9–Δex10}* and *Triobp^{Δex8/YHB226}* strains, we used a Zeiss 32-channel LSM880 laser-scanning confocal microscope equipped with an Airyscan detector and a 63×, 1.4-NA DICM27 Plan-Apochromat objective (Zeiss). Zen Black, v2.3, 64-bit acquisition software (Zeiss) was used to acquire raw data and perform Airyscan image processing. Settings for x–y pixel resolution, z-spacing, pinhole diameter, and grid selection were set according to software-suggested settings for optimal Nyquist-based resolution. Raw data were acquired using three-channel

(405-, 488-, and 561-nm) acquisition in frame mode; data processing for Airyscan-acquired images was performed using Zen Black software algorithm in auto mode. Maximum-intensity projections of Airyscan Z-stack images were obtained using Zen software. All images were acquired at room temperature using Zeiss 30°C microscope oil. For cochlea imaging, for each antibody, two to four images were acquired from two to four cochleae per genotype for each experiment, and experiments were repeated at least twice. Inner ear tissues from control and mutant littermates were stained and imaged using the same settings and on the same days to limit variability.

## Image analysis

To measure the dimensions of rootlet structures in separate experiments with different primary antibodies, we used the same Alexa Fluor 488 secondary antibody to ensure that PSFs were identical. For each rootlet measured, we used Fiji to crop a 20-pixel square (625 nm), collecting 20–50 examples in a single stack (16-bit image depth). All images in a stack were expanded to 500 pixels with bilinear interpolation, and the stack was averaged using Z Project Average and converted to 8-bit depth. For profile measurements, we drew a 15-pixel-wide transect across the center of the image, plotted that transect, and fitted it with a single Gaussian function. The FWHM is the side-to-side width of the Gaussian function measured at 50% of the peak of the function.

3D rendering of stereocilia and rootlets from Airyscan z-stack images was performed using Imaris (v9.5.0, Oxford Instruments). Background subtraction was performed in the image processing tab using a filter width of 1 μm for the phalloidin channel and 0.3 μm for the TRIOBP channel for all images. Volumetric surface areas of the stereocilia in each image were created from the phalloidin channel using creation parameters guided by the program and edited as needed so that each stereocilium was separately modeled. Volumetric surface areas of the rootlets in each image were then created using TRIOBP stain and edited as needed so that each rootlet was mapped correctly by a corresponding surface. Surface creation parameters for the stereocilia and rootlets were kept consistent across images. Stereocilia and rootlet surfaces were labeled as corresponding to row 1 or row 2 stereocilia and as corresponding to each cell in the image. To measure length and width of each surface, the object-oriented BoundingBoxOO Length C and A were used, respectively. The mean intensity of the TRIOBP signal within each rootlet surface was also measured. Stereocilia and rootlet measurements were averaged for each cell, and the CV for each cell was also calculated. Only cells with minimal overlap or collapse of the stereocilia were included in the analysis. Images were analyzed from two different experiments, with three to five images from two to three different cochlea used for each genotype for each experiment.

To analyze membrane association of ANKRD24 or TRIOBP-5 in HeLa cells, images of cells expressing either ANKRD24-GFP or GFP–TRIOBP-5 that were labeled with CF568 WGA were analyzed in Zen v2.6 (blue edition) software. For each cell, 10 profile scans 10 μm in length were drawn perpendicular to the plasma membrane, with the WGA peak oriented in the center of the

scan. Fluorescence intensities in both the WGA (568-nm) and GFP channels were recorded and normalized to the average baseline signals for each cell. Three cells each from two separate experiments were analyzed for each condition. To compare the signals at the plasma membrane, the average fluorescence intensities at the seven data points centered around the peak WGA signal were calculated for each profile scan.

To measure coaggregation of ANKRD24 and TRIOBP in Hela cells, we performed 3D surface rendering of all immunolabeled structures from SIM z-stack 63× images of HeLa cells using Imaris (v9.7.2). Background subtraction using a filter width of 2 µm and a 0.02-µm Gaussian filter were applied in the image processing tab for all channels in each image. Volumetric surface areas of the ANKRD24 and TRIOBP signals in each image were created using creation parameters guided by the program and edited as needed to surround the intracellular signals and minimize detection of background signals in untransfected cells. Surface creation parameters were kept consistent across images, with only the manual intensity threshold adjusted for each experiment date according to the average fluorescence intensity from that group of images. To minimize inclusion of background puncta and select for protein aggregates, only ANKRD24 surfaces >0.2 µm$^2$ in area and TRIOBP surfaces >0.05 µm$^3$ in volume were included in the analysis. We used the "overlapped volume ratio to surfaces" calculation for each TRIOBP surface, which measures the volume of the surface that overlaps with an ANKRD24 surface, divided by the total volume of the TRIOBP surface. Using this calculation, we measured the fraction of TRIOBP surfaces in each cell that had any overlap >0 with an ANKRD24 surface as well as the average overlapped volume ratio. For each construct being tested, ≥10 cells were analyzed from ≥6 images from ≥2 different experiments.

### EM

TEM used procedures similar to those described previously (Krey et al., 2018a). Inner ears from P6.5 or P21.5 *Ankrd24*$^{KO/+}$ or *Ankrd24*$^{KO/KO}$ mice were dissected in HBSS and immersion-fixed in a combination of 4% formaldehyde (1570; Electron Microscopy Sciences) and 1% glutaraldehyde (16020; Electron Microscopy Sciences) in 0.1 M phosphate buffer (pH 7.2) for 45 min to 1 h. The middle turns of the cochleae were dissected out and immersion-fixed on ice in 1% glutaraldehyde, 1% OsO$_4$ (19150; Electron Microscopy Sciences), and 0.1 M phosphate, pH 7.2, for 1 h, rinsed in water, and stained en bloc in 4% uranyl acetate for 30 min at room temperature in the dark. Organs were then rinsed in water (4 washes × 5 min), dehydrated in a graded series of acetone, and embedded in Embed-812 (14120; Electron Microscopy Sciences). Sections of 90 nm were collected on PELCO 200 mesh nickel grids (1GN200; Ted Pella), stained using UranyLess (22409; Electron Microscopy Sciences) and Reynold's lead citrate, and examined on an FEI Tecnai 12 BioTWIN transmission electron microscope (Thermo Fisher Scientific) operated at 80-kV (for P6.5 images) or 120-kV (for P21.5 images) accelerating voltage.

For scanning EM, periotic bones with cochleas were dissected in Leibovitz's L-15 medium (21083-027; Invitrogen) from P8.5 control and mutant littermates from *Ankrd24*$^{KO/+}$ × *Ankrd24*$^{KO/KO}$

crosses. An age-matched C57BL/6J control group was also included. Several small holes were made in periotic bones to provide access for fixative solutions; encapsulated cochleas were fixed for an hour in 2.5% glutaraldehyde in 0.1 M cacodylate buffer (15960; Electron Microscopy Sciences) supplemented with 2 mM CaCl$_2$. After washing with distilled water, the cochlear sensory epithelium was dissected out, and the tectorial membrane was lifted off manually. Cochlear tissues were dehydrated in a series of ethanol and critical-point dried using liquid CO$_2$ (EM CPD300; Leica). After immobilization on aluminum specimen holders using carbon tape, specimens were sputter-coated with 3–4 nm of platinum (EM ACE600; Leica). Samples were imaged using a Helios Nanolab 660 DualBeam Microscope (FEI).

### Coimmunoprecipitation

Expression of the constructs, coimmunoprecipitation, and protein immunoblots were carried out as described (Senften et al., 2006; Zhao et al., 2016; Liu et al., 2018). In brief, HEK293 cells (CRL-1573, RRID:CVCL_0045; ATCC) were cultured in DMEM supplemented with 10% heat-inactivated FBS (MT35011CV; Thermo Fisher Scientific) and 1% penicillin/streptomycin (MT30002CI; Thermo Fisher Scientific). Lipofectamine 2000 (11668019; Life Technologies) was used to transfect plasmids into HEK293 cells. 2 d after transfection, cells were harvested and lysed using radioimmunoprecipitation assay buffer containing 50 mM Hepes, 150 mM NaCl, 1% Triton X-100, 1 mM EDTA, and protease inhibitors. Then supernatant was collected after centrifugation at 13,000 rpm for 15 min at 4°C. Immunoprecipitations were then carried out using anti-HA agarose beads (E6779, RRID:AB_10109562; Sigma-Aldrich). After washing, immunoprecipitated proteins were eluted by boiling in SDS sample buffer.

### Functional tests for hearing and balance

DPOAE and VsEP experiments were carried out as described previously (Krey et al., 2016b). Mice were anesthetized by intraperitoneal injection of ketamine and xylazine (18 and 2 mg/ml; 5–7 µl/g body weight), followed by maintenance doses as needed to maintain adequate anesthesia. Core body temperature was maintained at 37.0 ± 0.2°C using a homeothermic heating pad (FHC). Recording electrodes were placed subcutaneously at the nuchal crest (noninverting electrode), behind the right pinna (inverting electrode), and at the hip (ground electrode). DPOAE stimuli were generated and controlled using TDT System III (RX6, PA5 modules) and SigGen/BioSig software. Pure tone frequencies ($f_2/f_1$ ratio = 1.25), at equal levels (L1 = L2 = 60 dB sound pressure level [SPL]), 150-ms duration, were generated by RX6 multifunction processor, attenuated through PA5 programmable attenuators, and routed through separate drivers to mix acoustically in the ear canal. Primary stimulus frequencies were such that the geometric mean, defined as GM = $(f_1 × f_2)^{0.5}$, had frequencies ranging from 5.8 to 47.6 kHz. Sounds were delivered from a closed-tube speculum sealed to the ear canal. Ear canal SPLs were recorded with a low-noise probe microphone (ER 10B+; Etymotic Research). The microphone output was amplified 10× and routed to the RX6 multifunction

processor for sampling at 100 kHz and averaging of fast Fourier transforms of the acoustic signals. The amplitudes of $f_1$, $f_2$, and the cubic ($2f_1 - f_2$) distortion product were measured from the fast Fourier transform waveform. The corresponding noise floor was determined from the average sound levels in the fifth and 10th frequency bins above and below the $2f_1 - f_2$ frequency bin.

For VsEP testing, linear acceleration ramps producing rectangular jerk pulses were generated and controlled using a National Instruments data acquisition system and custom software. Mice were placed supine on a stationary platform, and the head was secured within a spring clip coupled to a voltage-controlled mechanical shaker (model 132-2; Labworks). The head was oriented with nose up, and linear translation stimuli were presented in the naso-occipital axis parallel to the Earth-vertical axis. Vestibular stimuli consisted of 2-ms linear jerk pulses, delivered to the head using two stimulus polarities—normal, with an initial upward jerk, and inverted, with an initial downward jerk—at a rate of 17 pulses/s. Stimulus amplitudes ranged from +6 dB to –18 dB re: 1.0 g/ms (where 1 g = 9.8 m/s$^2$), adjusted in 3-dB steps. A broadband forward masker (50–50,000 Hz, 94 dB SPL) was presented during VsEP measurements to confirm the absence of auditory components (Jones and Jones, 1999). Signal averaging was used to extract the VsEP responses from the background electrophysiological activity. Ongoing electroencephalographic activity was amplified (200,000×), filtered (300–3,000 Hz, –6 dB points), and digitized beginning at the onset of each jerk stimulus (1,024 points, 10 μs/point) to produce one primary response trace. For each stimulus intensity and polarity, 128 primary responses were averaged to produce an averaged response waveform. Four averaged response waveforms were recorded for each stimulus intensity (two waveforms recorded for normal stimulus polarity and two for inverted polarity). Final individual response traces were produced by summing one averaged response to each stimulus polarity and dividing the result by two, thus producing two response traces for each stimulus intensity for each animal.

ABR and noise exposure experiments were carried out as described previously (Krey et al., 2016a). Animals were anesthetized with xylazine (10 mg/kg, intramuscular, IVX; Animal Health) and ketamine (40 mg/kg, intramuscular; Hospira) and placed on a heating pad in a sound-isolated chamber. Needle electrodes were placed subcutaneously near the test ear, both at the vertex and at the shoulder of the test ear side. A closed-tube sound-delivery system, sealed into the ear canal, was used to stimulate each ear. ABR measurements used tone bursts with a 1-ms rise time, applied at 4, 8, 12, 16, 24, and 32 kHz. Responses were obtained for each ear, and the tone-burst stimulus intensity was increased in steps of 5 dB. The threshold was defined as an evoked response of 0.2 μV from the electrodes.

A noise paradigm was used that produces both a temporary change in threshold, to examine reversible loss of sensitivity, and a partial permanent threshold change (Wilson et al., 2014). Mice were put into a small wire mesh divided cage, which was then placed into an open-field acoustic chamber. Mice were exposed to damaging levels of octave band limited noise for 2 h; the free field noise level was 0 (control) or 96 dB SPL (noise exposure) at 8–16 kHz with a 5-ms ramp-up in noise level. ABR measurements were taken before noise exposure, 1 h after noise exposure (defined as temporary threshold shift), and 2 wk after noise exposure (defined as permanent threshold shift). For all functional testing, mice of both sexes were used and the experimenters were aware of the animal genotypes during testing and analysis.

## MET from IHCs

Mice were decapitated, and inner ear tissue was dissected from P8–P12 mice of either sex and typically of unknown genotype. Organ of Corti tissues from the 5–12-kHz region were placed into a recording chamber as previously described (Peng et al., 2013). Hair cells were imaged on a BX51 upright fixed-stage microscope (Olympus) using a 100×, 1.0-NA dipping lens. The dissection and extracellular solution contained (in mM) 140 NaCl, 2 CaCl$_2$, 0.5 MgCl$_2$, 10 Hepes, 2 Na-ascorbate, 2 Na-pyruvate, and 6 dextrose. The osmolality was 300–310 mOsm, and pH was 7.4. The tectorial membrane was peeled off before mounting in the dish. The recording chamber was placed onto the stage, and apical perfusion and bath perfusion were added to the dish with the same solution as described. After a whole-cell recording was obtained, the apical perfusion was turned off to limit additional mechanical stimulation of the hair bundle or disruption of fluid jet flow.

Whole-cell patch recordings were obtained using thick-walled borosilicate pipettes with electrode resistances of 3–5 MΩ and tip size 1.5–2.2 μm inner diameter, pulled on a Sutter P95 micropipette puller. The internal solution contained (in mM) 100 CsCl, 30 ascorbate, 3 Na$_2$ATP, 5 phosphocreatine, 10 Hepes, and 1 Cs$_4$BAPTA; osmolality was 290 mOsm, and pH was 7.2. An Axopatch 200b amplifier (Molecular Devices) coupled to an Iotech 3000 data acquisition board (and PC computer) were used for all measurements. Data were sampled at 100 kHz and filtered with an 8-pole Bessel filter at 10 kHz. Junction potentials were estimated at 4 mV and corrected offline. Uncompensated series resistance was 8.7 ± 4 MΩ ($n$ = 30), and whole-cell capacitance was 9.7 ± 1.4 pF ($n$ = 30). Cells were held at –80 mV for all experiments, and calcium currents were used as a quality control test for the recording. No difference was found in calcium current properties between genotypes. Cells were included only if recordings remained stable throughout the timeframe of data capture.

Hair bundles were stimulated with a fluid jet driven by a piezo electric disc bender 592 (27 mm, 4.6 kHz; 7BB-27-4L0; Murata Electronics). Discs were mounted in a 3D-printed housing to minimize fluid volume being moved by the disc. Thin-walled borosilicate glass was used to deliver fluid to the bundle. Tip sizes of 8–10-μm diameter were selected, as they uniformly stimulate IHC bundles when placed 1–3 μm from the bundle face (Peng et al., 2021). Three cycles of a 40-Hz sine wave were used to activate MET channels. Voltage was varied to the disc bender to maximize current amplitudes. Maximal current was identified by flattening of the peak response when channels were opened. The damage protocol was empirically determined. A 167-Hz stimulus set at 125% of maximal stimulus had a small but typically reversible effect on MET current amplitudes in control animals. Higher amplitudes resulted in damage like that

described for the mutants in terms of asynchronous bundle motion and permanently reduced currents. Motion to a constant stimulus was measured during the experiment and scaled across stimulus amplitudes with the assumption of linearity with voltage. For these experiments, we chose only those hair cells that had two rows of thick stereocilia, not the more aberrant bundles highlighted in Fig. 6.

MET current maximums were measured as the current difference between maximal on and maximal off stimulations. Current–displacement (IX) plots were generated by plotting the current from the 40-Hz stimulus against the current response for the first half-cycle of stimulation that opened channels. Given the hysteresis in hair bundle response to sinusoidal stimulation, we analyzed only the first cycle before any hysteresis response. Plots were fitted with a two-state Boltzmann function (Corey and Hudspeth, 1983) of the following form:

$$Y = y_0 + \left[ I_{max} / \left\{ 1 + e^{[(x_0 - x)^* z]} \right\} \right],$$

where $y_0$ is the $y$-intercept, $I_{max}$ is the maximal current, $X_0$ is the set point, and $z$ is the slope. The percent damage was calculated as

$$100 \times \left[ 1 - \left( I_{maxpre} - I_{maxpost} \right) / I_{maxpre} \right],$$

where pre and post represent responses before and after damage, respectively.

### Statistics
Unless otherwise stated, statistical comparisons between two sets of data used the two-sided Student's $t$ test with unpaired data and the untested assumption of equal variance. In figures, P values are indicated by asterisks: *, $P < 0.05$; **, $P < 0.01$; ***, $P < 0.001$; and ****, $P < 0.0001$.

For Fig. 2 J, we carried out one-sample $t$ tests to determine whether the mean of each experimental population was statistically different from 1.00 (no membrane enrichment). For these data, P values were as follows: GFP signal in ANKRD24-GFP transfection, <0.0001; WGA signal in ANKRD24-GFP transfection, <0.0001; GFP signal in GFP–TRIOBP-5 transfection, 0.0599; TRIOBP-5 WGA signal in GFP–TRIOBP-5 transfection, <0.0001.

For Fig. 3, G and H, we carried out $t$ tests to determine whether TRIOBP-1 or TRIOBP-4 overlap fraction (Fig. 4 G) or overlap ratio (Fig. 4 H) relative to ANKRD24 was significantly different from that of TRIOBP-5. For these data, P values were as follows: overlap fraction: TRIOBP-1, 0.407; TRIOBP-1, 0.0269; overlap ratio: TRIOBP-1, 0.758; TRIOBP-1, 0.0327.

For Fig. 4, we carried out $t$ tests to determine whether the deletion and truncation construct overlap fraction (Fig. 4 G) or overlap ratio (Fig. 4 H) relative to TRIOBP-5 was significantly different from that of ANKRD24-HA (as all deletion and truncation constructs were C-terminally HA-tagged). Overlap fraction: ΔN, 0.635; ΔANK, 0.329; ΔAla, 0.020; Trunc2, 0.00052; Trunc3, 0.0425; Trunc4, 0.00165; Trunc5, 0.00116; and Trunc6, 0.838. Overlap ratio: ΔN, 0.782; ΔANK, 0.123; ΔAla, 0.986; Trunc2, <0.0001; Trunc3, <0.0001; Trunc4, 0.223; Trunc5, <0.0001; and Trunc6, 0.936.

For Fig. 5 F, we measured ABRs from 16 ears from 8 animals (5 female, 3 male) for C57BL/6, 28 ears from 14 animals (9

female, 5 male) for $Ankrd24^{KO/+}$, and 29 ears from 15 animals (7 female, 8 male) for $Ankrd24^{KO/KO}$. For Fig. 5 G, we measured ABRs from 20 ears from 10 animals (5 female, 5 male) for C57BL/6, 16 ears from 8 animals (6 female, 2 male) for $Ankrd24^{KO/+}$, and 16 ears from 8 animals (2 female, 6 male) for $Ankrd24^{KO/KO}$. Because they were right-censored at 80 dB, the ABR data in Figs. 5, F and G, and Fig. 9 were analyzed by Cox proportional hazard model. This approach is akin to use the generalized estimating equation model (Liang and Zeger, 1986) for noncensored and repeated-measures or correlated data (Therneau and Grambsch, 2000). The model fitting was performed by the rms (https://CRAN.R-project.org/package=rms) and survival (Therneau and Grambsch, 2000) packages of R Statistical Computing Environment (https://www.R-project.org/). For Fig. 5 F, P values were as follows: 4 kHz, 0.992 for KO-B6 ($Ankrd24^{KO/KO}$ to $Ankrd24^{KO/+}$) and 0.0006 for KO-Het ($Ankrd24^{KO/KO}$ to C57BL/6); 8 kHz, 0.8112 for KO-B6 and 0.0080 for KO-Het; 12 kHz, 0.0374 for KO-B6 and <0.0001 for KO-Het; 16 kHz, <0.0001 for KO-B6 and <0.0001 for KO-Het; 24 kHz, <0.0001 for KO-B6 and <0.0001 for KO-Het; and 32 kHz, <0.0001 for KO-B6 and <0.0001 for KO-Het. For Fig. 5 G, P values were as follows: 4 kHz, 0.0007 for KO-B6 and 0.0007 for KO-Het; 8 kHz, 0.0027 for KO-B6 and 0.0082 for KO-Het; 12 kHz, 0.0002 for KO-B6 and 0072 for KO-Het; 16 kHz, 0.6300 for KO-B6 and 0.6482 for KO-Het; 24 kHz, 0.7496 for KO-B6 and 0.7476 for KO-Het; and 32 kHz, 0.8778 for KO-B6 and 0.8738 for KO-Het.

For Figs. 5, H and I, we measured DPOAEs and VsEPs from four animals for +/+ (two female, two male), four animals for $Ankrd24^{KO/+}$ (two female, two male), and four animals for $Ankrd24^{KO/KO}$ (one female, three male). For Fig. 5 H, a paired $t$ test was used to test the difference between DPOAEs and the corresponding noise floors in each genotype at a frequency. A one-way ANOVA model was used to test the differences of DPOAEs from noise floor among genotypes at each frequency. For Fig. 5 I, a one-way ANOVA model was used to test differences of VsEPs across genotypes. The model yielded P values of 0.288 for the $Ankrd24^{KO/KO}$ to $Ankrd24^{KO/+}$ comparison and 0.068 for the $Ankrd24^{KO/KO}$ to +/+ comparison.

For Fig. 9, A–E, we used 16 ears from 8 animals (5 female, 3 male) for C57BL/6 for each experimental group, 28 ears from 14 animals (9 female, 5 male) for the $Ankrd24^{KO/+}$ control group, 28 ears from 14 animals (6 female, 8 male) for the $Ankrd24^{KO/+}$ noise exposure group, 30 ears from 15 animals (7 female, 8 male) for the $Ankrd24^{KO/KO}$ control group, and 28 ears from 14 animals (8 female, 6 male) for the $Ankrd24^{KO/KO}$ noise exposure group; 3 $Ankrd24^{KO/+}$ animals and 3 $Ankrd24^{KO/KO}$ animals died during ABR acquisition and were not included in the dataset. In the analysis of these data, clustering information from multiple measures per ear was included with other covariates in the Cox model. For the baseline-minus-2-wk ABR difference analysis of Fig. 9, G–I, P values were as follows: 4 kHz: 0.297 for B6 noise to B6 control, 0.191 for Het noise to Het control, 0.013 for KO noise to KO control, 0.002 for KO noise to B6 noise, 0.004 for KO noise to Het noise, 0.033 for KO control to B6 control, 0.334 for KO control to Het control; 8 kHz: 0.349 for B6 noise to B6 control, 0.591 for Het noise to Het control, 0.325 for KO noise to KO control, 0.179 for KO noise to B6 noise, 0.479 for KO noise to Het

noise, 0.102 for KO control to B6 control, 0.436 for KO control to Het control; 12 kHz: 0.164 for B6 noise to B6 control, 0.306 for Het noise to Het control, 0.008 for KO noise to KO control, 0.082 for KO noise to B6 noise, 0.032 for KO noise to Het noise, 0.458 for KO control to B6 control, 0.226 for KO control to Het control; 16 kHz: 0.432 for B6 noise to B6 control, 0.863 for Het noise to Het control, 0.018 for KO noise to KO control, 0.150 for KO noise to B6 noise, 0.308 for KO noise to Het noise, 0.672 for KO control to B6 control, 0.274 for KO control to Het control; 24 kHz: 0.081 for B6 noise to B6 control, 0.620 for Het noise to Het control, 0.801 for KO noise to KO control, 0.302 for KO noise to B6 noise, 0.928 for KO noise to Het noise, 0.091 for KO control to B6 control, 0.960 for KO control to Het control; and 32 kHz: 0.0003 for B6 noise to B6 control, 0.345 for Het noise to Het control, 0.006 for KO noise to KO control, 0.577 for KO noise to B6 noise, 0.877 for KO noise to Het noise, <0.0001 for KO control to B6 control, <0.0001 for KO control to Het control. In Fig. 9, G–I, only P value symbols for the noise-control within-genotype comparisons are indicated. For Fig. 9 P, we examined 28 $Ankrd24^{KO/+}$ IHCs and 34 $Ankrd24^{KO/KO}$ IHCs; these two distributions were significantly different at P < 0.0001.

### Online supplemental material
Five figures are included in the supplemental material: Fig. S1 shows localization of ANKRD24 at stereocilia insertions and along rootlets. Fig. S2 shows ANKRD24 and TRIOBP expression in HeLa cells. Fig. S3 shows ANKRD24 deletion constructs in HeLa cells. Fig. S4 shows ANKRD24 antibody characterization. Fig. S5 shows TRIOBP in $Ankrd24^{KO/KO}$ hair cells.

## Acknowledgments
We thank Merle Gilbert PhD for initiating this project, Elizabeth Bernhard and Sherly Michel for genotyping of the *Triobp* mutant strain mice, Patrick Deers for help with time-sensitive requests of tail snips and tattoos of *Triobp* neonatal pups, and Amelia Baumgardner for assistance with DPOAE and VsEP experiments.

We received support from the following core facilities: mass spectrometry from the OHSU Proteomics Shared Resource (partial support from NIH core grants P30 EY010572 and P30 CA069533; Orbitrap Fusion S10 OD012246), mouse production from the OHSU Transgenic Mouse Models core, confocal microscopy from the OHSU Advanced Light Microscopy Core @ The Jungers Center (P30 NS061800 provided support for imaging), and electron microscopy from the OHSU Multiscale Microscopy Core. PGBG was supported by NIH grants R01DC002368 and R01DC014427. This study was also supported (in part) by the Intramural Research Programs of the NIH, NIDCD DC000039 to T.B. Friedman; B. Zhao was supported by NIH grant R01DC017147, while A.J. Ricci and J. Kim were supported by NIH R01DC0003896.

The authors declare no competing interests.

Author contributions: Conceptualization—P.G. Barr-Gillespie, J.F. Krey, I.A. Belyantseva. Data curation—J.F. Krey, C. Liu, A.J. Ricci. Formal analysis—P.G. Barr-Gillespie, J.F. Krey, A.J. Ricci, D. Choi, A.L. Nuttall, S.M. Jones. Funding acquisition—P.G. Barr-Gillespie, T.B. Friedman, A.J. Ricci, B. Zhao, A.L. Nuttall.

Investigation—J.F. Krey, I.A. Belyantseva, A.J. Ricci, C. Liu, S.M. Jones, J. Kim, S. Foster, M. Bateschell, R.A. Dumont, P. Chatterjee, R.S. Morrill. Methodology—P.G. Barr-Gillespie, J.F. Krey, I.A. Belyantseva, A.J. Ricci, C. Liu, S.M. Jones, D. Choi, B. Zhao. Project administration—P.G. Barr-Gillespie. Resources—P.G. Barr-Gillespie, T.B. Friedman, A.J. Ricci, B. Zhao, S.M. Jones, L.M. Fedorov, J. Goldsmith. Software—n/a. Supervision—P.G. Barr-Gillespie, T.B. Friedman, B. Zhao. Validation—J.F. Krey (ANKRD24 antibody validation), I.A. Belyantseva (Triobp-1/5 antibody validation). Visualization—P.G. Barr-Gillespie, J.F. Krey, I.A. Belyantseva, A.J. Ricci, B. Zhao. Writing (original draft)—P.G. Barr-Gillespie and J.F. Krey. Writing (review & editing)—all authors.

Submitted: 29 September 2021

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

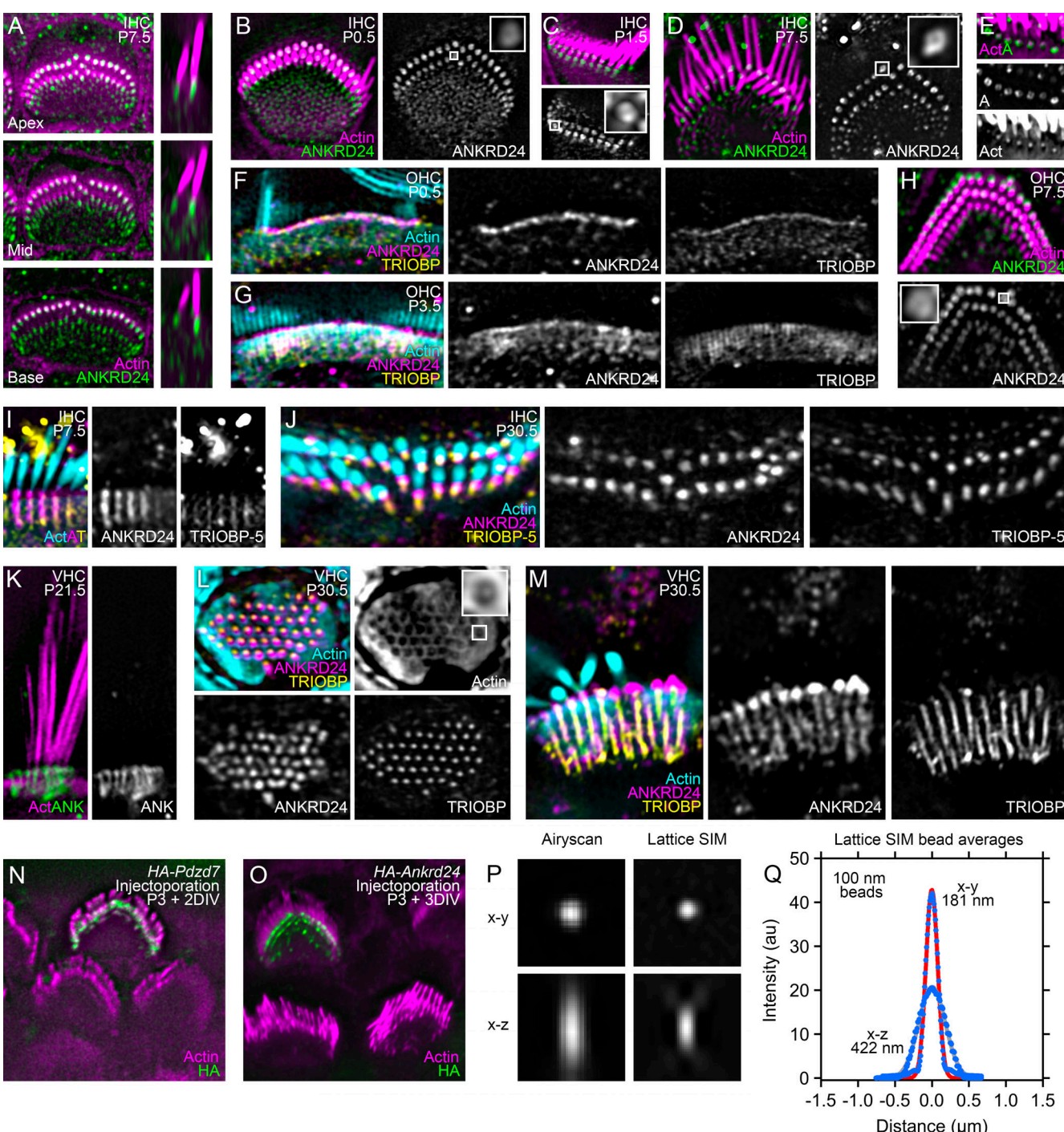

Figure S1. **Localization of ANKRD24 at stereocilia insertions and along rootlets.** All images used lattice SIM imaging. **(A)** Apical to basal ANKRD24 expression in IHCs on P7.5. Left, surface view; right, reslice. **(B–E)** ANKRD24 expression in midcochlea IHCs; insets show rings at stereocilia insertion. Ages are P0.5 (B), P1.5 (C), and P7.5 (D and E). **(F)** ANKRD24 and TRIOBP expression at stereocilia insertion in P0.5 mid-OHC. Rootlets have not developed yet. **(G)** ANKRD24 and TRIOBP expression at stereocilia insertion and rootlets in P3.5 basal OHC. Rootlets have developed, and TRIOBP labeling marks them. **(H)** ANKRD24 rings are apparent (inset) in P7.5 mid-OHC. **(I and J)** Labeling with TRIOBP-5-selective antibody. Basal IHC on P7.5 (I) and apical IHC on P30.5 (J). **(K)** VHC in profile (P21.5). **(L and M)** Utricular hair cell fragments isolated by adherence to glass. Isolated cuticular plate (L) and cuticular plate with stereocilia in profile (M). The contrast of the inset in the actin panel in L was enhanced to more clearly show the rootlet and clear region. **(N and O)** Localization of injectoporated HA-PDZD7 (N) and HA-ANKRD24 (O) in OHCs. **(P)** Images of single 100-nm Tetraspeck beads (488-nm illumination) using Airyscan (left) or lattice SIM imaging. Images are either the x–y stack average-projected in z (top) or x–z reslice projected in y. **(Q)** Plots of x–y and x–z profiles using stack derived from the average of 147 beads. Zeiss Zen Black software calculated FWHM values for appropriate PSFs of red channel x–y, 161 nm; red channel x–z, 521 nm; green channel x–y, 154 nm; green channel x–z, 433 nm. Panel full widths: A, 10 µm (left) and 3 µm (right); B, 8 µm (inset, 0.5 µm); C and D, 10 µm (insets, 1 µm); E, 4 µm; F and G, 8 µm; H, 6 µm (inset, 0.4 µm); I, 3 µm; J, 7.4 µm; K, 5 µm; L, 5 µm (inset, 0.5 µm); M, 5.5 µm; N, 1.47 µm.

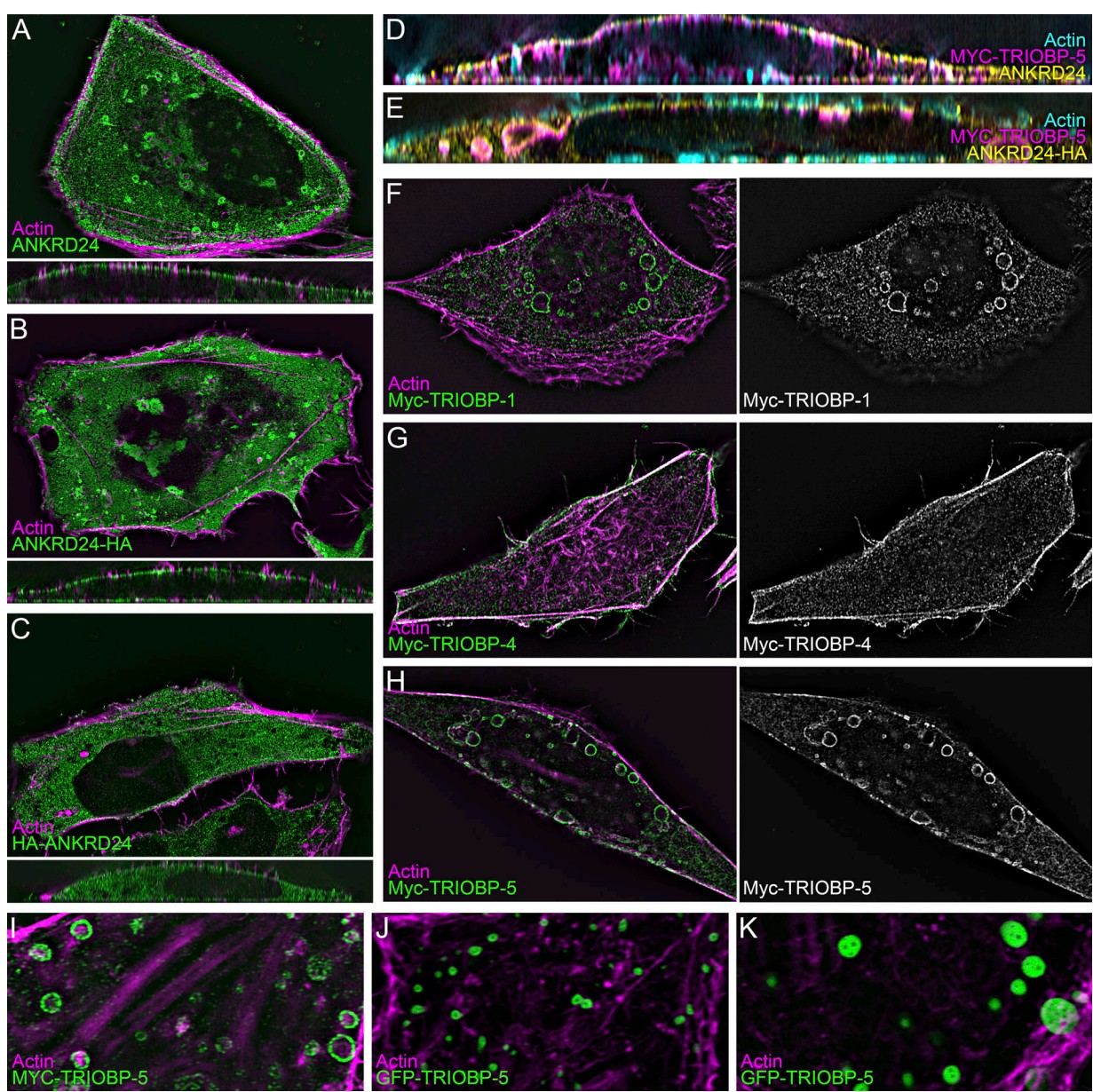

Figure S2. **ANKRD24 and TRIOBP expression in HeLa cells.** All images used lattice SIM imaging. **(A–C)** Impact of tag location on ANKRD24 localization. Images of cells transfected with untagged ANKRD24 (A), ANKRD24 tagged with HA on the C-terminus (ANKRD24-HA; B), or ANKRD24 tagged with HA on the N-terminus (HA-ANKRD24; C). ANKRD24-HA had a labeling pattern very similar to that of untagged ANKRD24. By contrast, unlike the other constructs, HA-ANKRD24 did not target to the membrane. **(D and E)** Both untagged ANKRD24 (D) and ANKRD24-HA (E) colocalize with Myc–TRIOBP-5. **(F–H)** Localization of TRIOBP splice forms in the absence of ANKRD24. **(I–K)** Variability of the size and number of TRIOBP-5 aggregates in different transfected cells. Panel full widths: A–C (including insets), 60 µm; D–E, 55 µm; F–H, 60 µm; I–K, 20 µm.

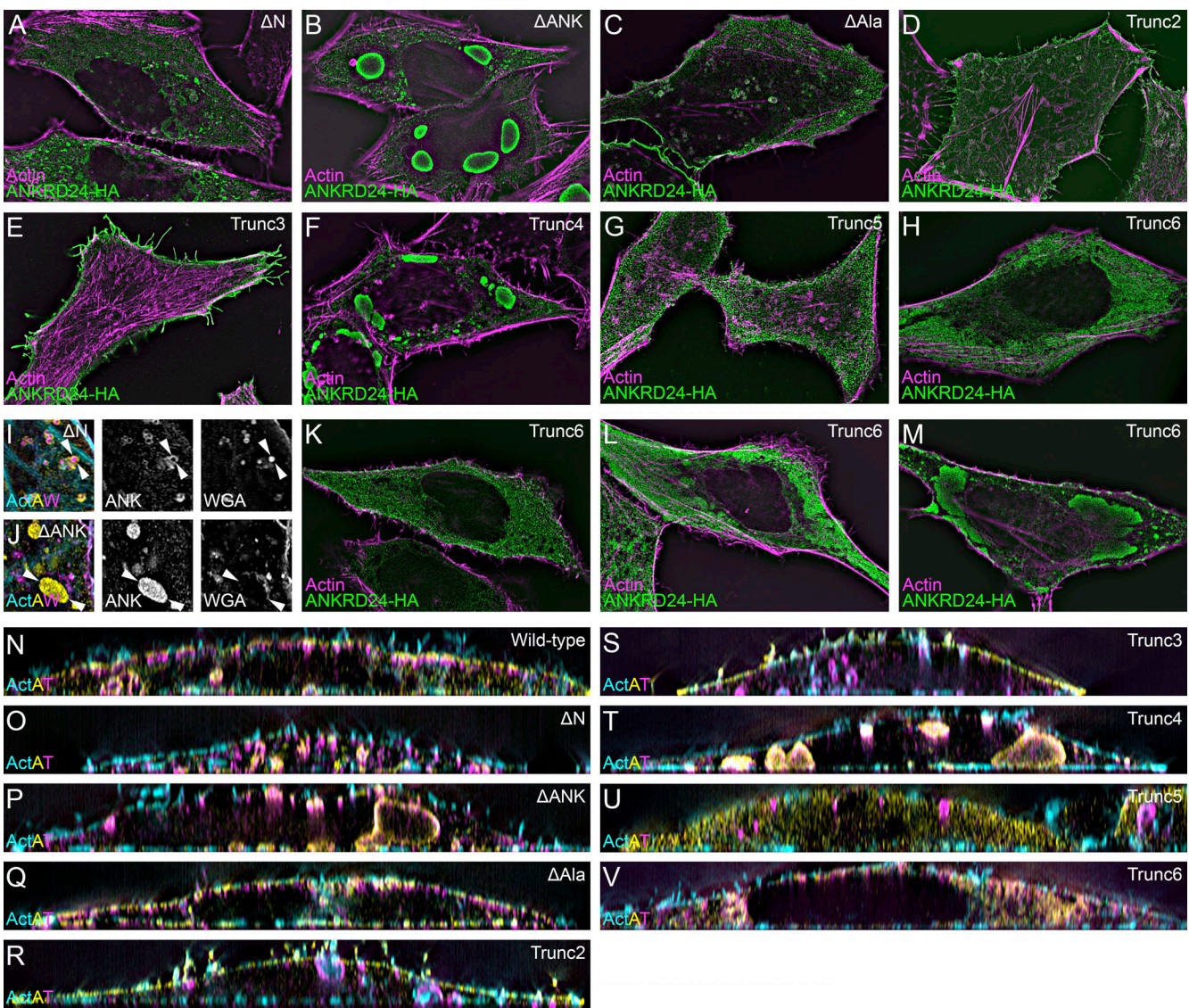

Figure S3. **ANKRD24 deletion constructs in HeLa cells.** All images used lattice SIM imaging. **(A–H)** ANKRD24 deletion constructs in the absence of TRIOBP cotransfection; patterns are similar to those seen when deletion constructs are coexpressed with TRIOBP (Fig. 4). **(I)** WGA staining material is sequestered in the core of ΔN-ANKRD24 aggregates. ActAW refers to cyan actin labeling, yellow ANKRD24 labeling, and magenta WGA labeling. **(J)** WGA staining material is absent from ΔANK-ANKRD24 aggregates. Arrowheads in I and J indicate examples of aggregated ANKRD24. **(K–M)** Variable ANKRD24 labeling pattern in cells transfected with Trunc6-HA. The labeling pattern ranged from relatively cytoplasmic (K) through large aggregates (M). **(N–V)** Reslice views of cells cotransfected with ANKRD24 deletion constructs. ActAT refers to cyan actin labeling, yellow ANKRD24 labeling, and magenta TRIOBP-5 labeling. Panel full widths: A–H and K–V, 60 μm; I and J, 9.9 μm.

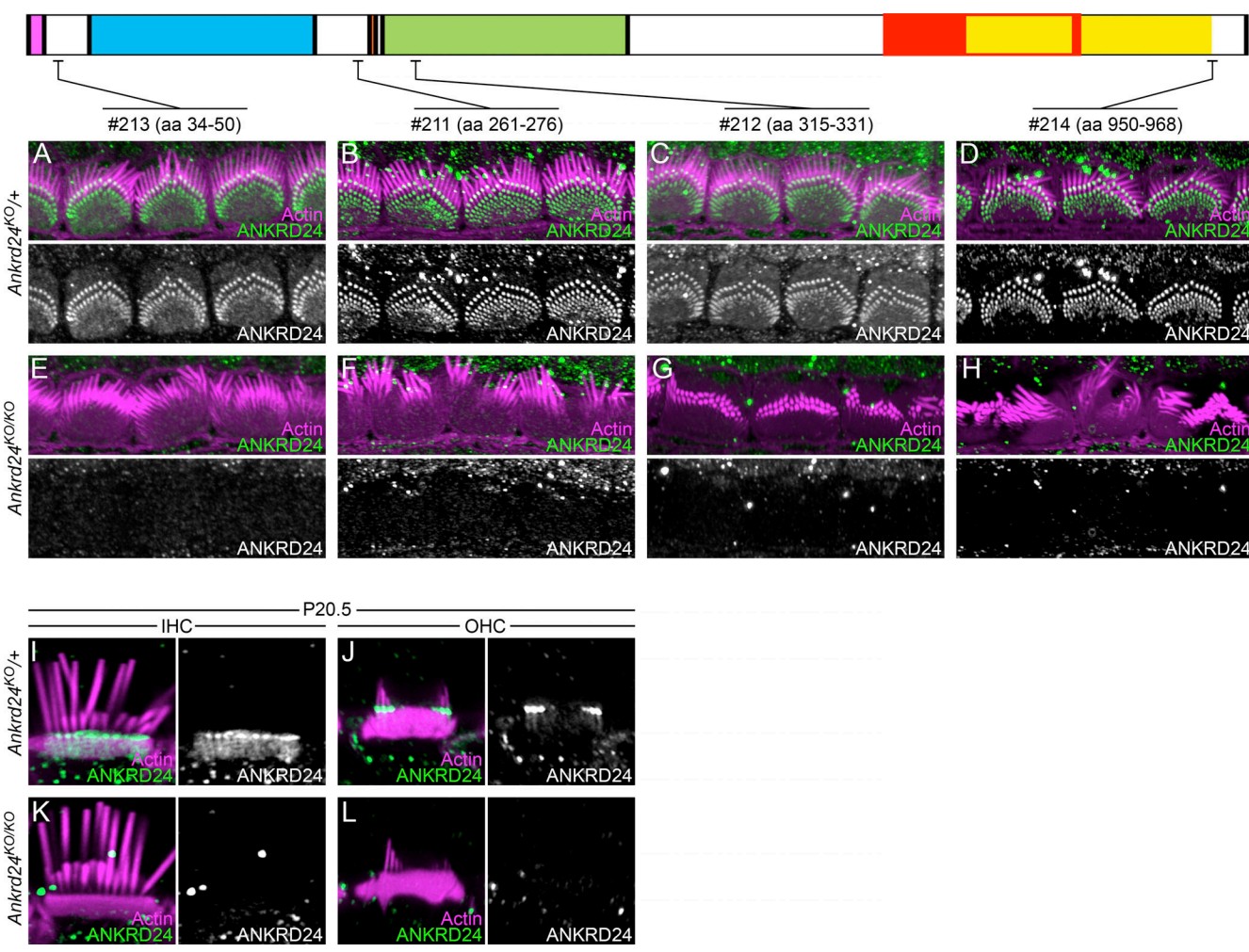

Figure S4.  **ANKRD24 antibody characterization. (A–H)** Characterization of four rabbit anti-ANKRD24 peptide antibodies on P7.5. The top diagram shows the location of the peptides used for antibody production. A–D, localization in *Ankrd24^{KO/+}* heterozygous mice; E–H, localization in *Ankrd24^{KO/KO}* knockout mice. Note that antibody no. 213 targets an epitope that is located before the genomic deletion; E confirms that *Ankrd24^{KO/KO}* is effectively a null. **(I–L)** ANKRD24 labeling with no. 213 rabbit anti-ANKRD24 antibody in IHCs (I and K) and OHCs (J and L) on P20.5. I and J, localization in *Ankrd24^{KO/+}* heterozygous hair cells; K and L, localization in *Ankrd24^{KO/KO}* knockout hair cells. Panel full widths: A–H, 30 µm; I–L, 10 µm.

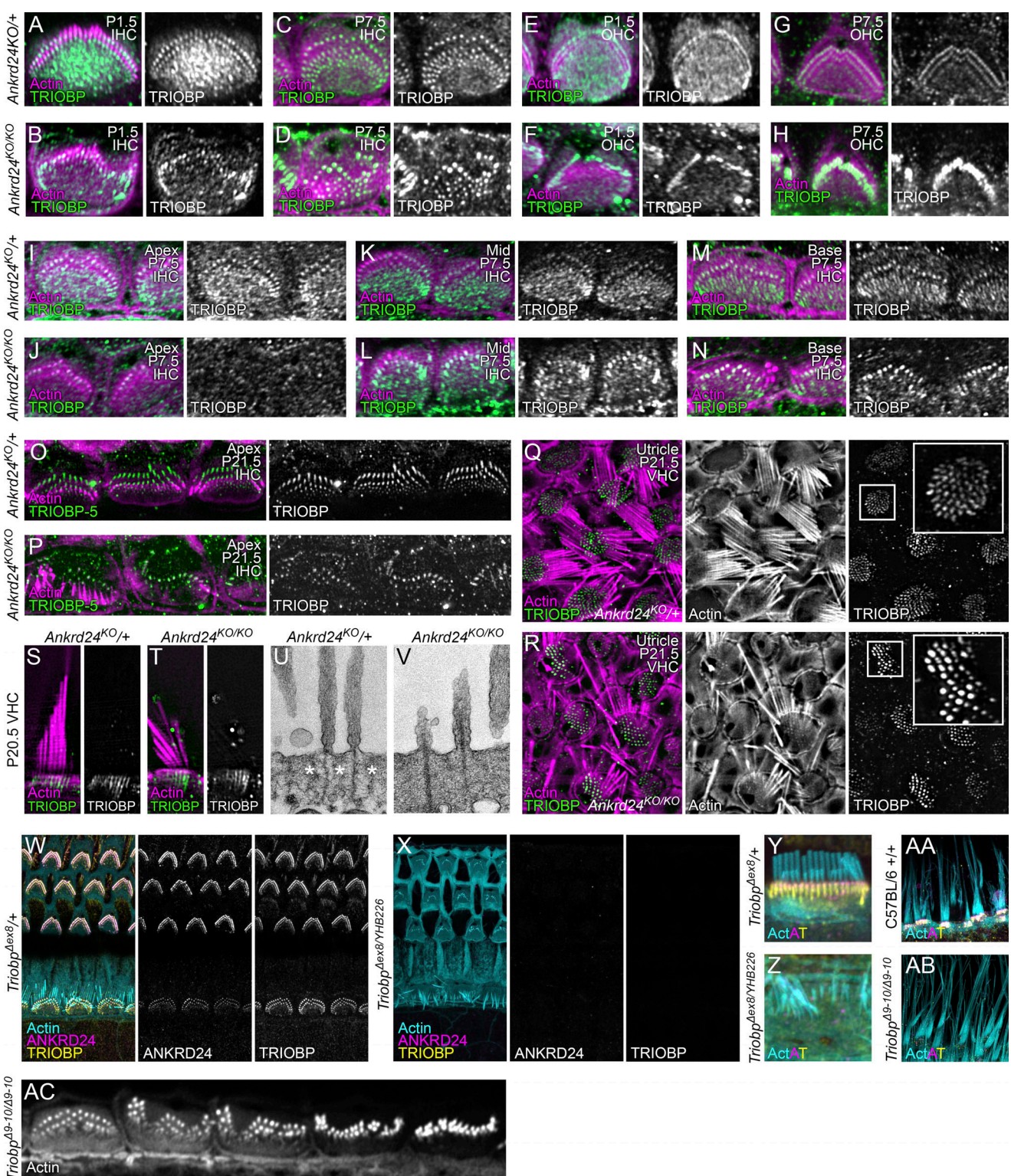

Figure S5. **TRIOBP in *Ankrd24^{KO/KO}* hair cells. (A–H)** F-actin and TRIOBP in IHCs and OHCs from *Ankrd24^{KO/+}* and *Ankrd24^{KO/KO}* mice on P1.5 and P7.5. **(I–N)** Actin and TRIOBP in IHCs from *Ankrd24^{KO/+}* and *Ankrd24^{KO/KO}* P7.5 mice at the cochlea apex, middle, and base. **(O and P)** F-actin and TRIOBP-5 in IHCs from *Ankrd24^{KO/+}* and *Ankrd24^{KO/KO}* mice on P21.5. **(Q and R)** Actin and TRIOBP in utricle from *Ankrd24^{KO/+}* and *Ankrd24^{KO/KO}* mice on P21.5. **(S and T)** F-actin and TRIOBP in VHCs of utricle from *Ankrd24^{KO/+}* and *Ankrd24^{KO/KO}* mice on P20.5. **(U and V)** TEM images of VHCs in utricles of *Ankrd24^{KO/+}* and *Ankrd24^{KO/KO}* mice on P21.5. Asterisks in U bracket the clear channel surrounding the rootlet. **(W–Z)** ANKRD24 is present in rootlets of *Triobp^{Δ8/+}* control cochlear hair cells on P17 (W and Y) and is absent from *Triobp^{Δ8/YHB226}* compound heterozygote cochlear hair cells on P17 (X and Z). **(AA and AB)** At P12, ANKRD24 is present in rootlets of C57BL/6 control VHCs (AA) and is absent from *Triobp^{Δ9–Δ10/Δ9–Δ10}* knockout VHCs (AB). **(AC)** At P10, warped bundles are seen in *Triobp^{Δ9–Δ10/Δ9–Δ10}* knockout IHCs. Superresolution modality: A–N and W–AC, Airyscan; O–T, lattice SIM. Panel full widths: A–H, 10 µm; I–N, 16 µm; O and P, 24 µm; Q and R, 17.5 µm; S and T, 5 µm; U and V, 1 µm; W and X, 30 µm; Y and Z, 8 µm; AA and AB, 30 µm; AC, 40 µm.

