## [Peer Review File · The Journal of Cell Biology]

ANKRD24 organizes TRIOBP to reinforce stereocilia insertion points

Jocelyn Krey, Chang Liu, Inna Belyantseva, Michael Bateschell, Rachel Dumont, Jennifer Goldsmith, Paroma Chatterjee, Rachel Morrill, Lev Fedorov, Sarah Foster, Jinkyung Kim, Alfred Nuttall, Sherri Jones, Dongseok Choi, Thomas Friedman, Anthony Ricci, Bo Zhao, and Peter Barr-Gillespie

Corresponding Author(s): Peter Barr-Gillespie, Oregon Health & Science University

Review Timeline:	Submission Date:	2021-09-29
	Editorial Decision:	2021-11-18
	Revision Received:	2022-01-07
	Editorial Decision:	2022-01-13
	Revision Received:	2022-01-14

Monitoring Editor: John Wallingford

Scientific Editor: Andrea Marat

Transaction Report:

DOI: <https://doi.org/10.1083/jcb.202109134>

November 18, 2021

Re: JCB manuscript #202109134

Dr. Peter Barr-Gillespie
Oregon Health & Science University
Oregon Hearing Research Center
L335A
3181 SW Sam Jackson Pk Rd
Portland, OR 97239

Dear Dr. Barr-Gillespie,

Thank you for submitting your manuscript entitled "ANKRD24 organizes TRIOBP to reinforce stereocilia insertion points". The manuscript was assessed by expert reviewers, whose comments are appended to this letter. We invite you to submit a revision if you can address the reviewers' key concerns, as outlined here.

As you will see, both reviewers are overall positive regarding your study. We agree that for resubmission to JCB all requested quantifications are essential. However, while reviewer #1 has provided several interesting suggestions of points to investigate, as these are not essential for your main conclusions providing experimental data to address them is optional.

GENERAL GUIDELINES:

Text limits: Character count for an Article is < 40,000, not including spaces. Count includes title page, abstract, introduction, results, discussion, acknowledgments, and figure legends. Count does not include materials and methods, references, tables, or supplemental legends.

Figures: Articles may have up to 10 main text figures. Figures must be prepared according to the policies outlined in our Instructions to Authors, under Data Presentation, <https://jcb.rupress.org/site/misc/ifora.xhtml>. All figures in accepted manuscripts will be screened prior to publication.

Supplemental information: There are strict limits on the allowable amount of supplemental data. Articles may have up to 5 supplemental figures. Up to 10 supplemental videos or flash animations are allowed. A summary of all supplemental material should appear at the end of the Materials and methods section.

Please note that JCB now requires authors to submit Source Data used to generate figures containing gels and Western blots with all revised manuscripts. This Source Data consists of fully uncropped and unprocessed images for each gel/blot displayed in the main and supplemental figures. Since your paper includes cropped gel and/or blot images, please be sure to provide one Source Data file for each figure that contains gels and/or blots along with your revised manuscript files. File names for Source Data figures should be alphanumeric without any spaces or special characters (i.e., SourceDataF#, where F# refers to the associated main figure number or SourceDataFS# for those associated with Supplementary figures). The lanes of the gels/blots should be labeled as they are in the associated figure, the place where cropping was applied should be marked (with a box), and molecular weight/size standards should be labeled wherever possible.

As you may know, the typical timeframe for revisions is three to four months. However, we at JCB realize that the implementation of social distancing and shelter in place measures that limit spread of COVID-19 also pose challenges to scientific researchers. Lab closures especially are preventing scientists from conducting experiments to further their research. Therefore, JCB has waived the revision time limit. We recommend that you reach out to the editors once your lab has reopened

to decide on an appropriate time frame for resubmission. Please note that papers are generally considered through only one revision cycle, so any revised manuscript will likely be either accepted or rejected.

Thank you for this interesting contribution to Journal of Cell Biology. You can contact us at the journal office with any questions, cellbio@rockefeller.edu or call (212) 327-8588.

Sincerely,

John Wallingford, PhD
Monitoring Editor

Andrea L. Marat, PhD
Senior Scientific Editor

Journal of Cell Biology

Reviewer #1 (Comments to the Authors (Required)):

In this manuscript by Krey et al., the authors report an essential role of the ankrin-repeat protein ANKRD24 in structural stability of the hair cell stereocilia through extensive functional and biochemical analysis. Using super-resolution light microscopy and structure-function analysis in heterologous cells, they show that ANKRD24 localizes to the stereocilium insertion point and lower rootlet, interacts with the rootlet actin regulator TRIOBP and they mutually regulate each other's localization. ANKRD24 knockout mice have misshapen hair bundles (HB), progressive hearing loss and vulnerability to noise damage. Overall, this is a thorough and rigorous analysis that convincingly establishes ANKRD24 as a key structural component of the stereocilium rootlet crucial for HB stability and hearing.

There are several issues that require clarification and/or further interrogation, as detailed below:

1. It is not clear how ANKRD24 is localized to the stereocilium insertion point. There is no evidence that the N-terminal membrane-interacting region of ANKRD24 is sensitive to membrane curvature. Another possibility worth considering and investigating is that ANKRD24 interacts with specific lipids enriched in the previously defined "taper domain" (Zhao et al., 2012).
2. Related to above, it is also possible that ANKRD24-membrane interaction is allosterically regulated by ANKRD24-TRIOBP interaction, as suggested by the predicted intramolecular interaction (Line 357). This should be discussed and investigated by more carefully examining ANKRD24 localization relative to cell membranes in TRIOBP5 mutants at multiple developmental stages, in addition to data shown in Figure 8P-S. Moreover, it would be informative to investigate the localization of ANKRD24-ANK and Trunc2 using the injectoporation assay, to shed light on this potential regulation.
3. It appears that the area of the fonticulus in the ANKRD24 mutant hair cells is aberrantly enlarged (Figure 7P-W, more apparent in OHCs), suggesting possible "erosion" of the cuticular plate. This defect should be quantified and discussed.
4. The warped HB morphology defects in ANKRD24 mutants are not seen in TRIOBP mutants (Kitajiri et al., 2010), suggesting that ANKRD24 and TRIOBP have separate functions during HB morphogenesis. This should be discussed.
5. Given the progressive hearing loss and susceptibility to noise damage in ANKRD24 mutants, HB morphology should be assessed by SEM in older adults and following noise exposure.

Specific comments:

1. The max-intensity projection images shown in Figures 2-4 do not convincingly demonstrate co-localization. Quantifications of Pearson's coefficient should be performed using single-plane images.
2. The VsEP (and DPOAE) data in Figure 5 were based on small n's and not yet significant ($p=0.068$). Suggest either increasing n's to reach statistical significance or remove from the main Figure.
3. Figure 6J showed IHCs with normal HB morphology, whereas Figure 8G showed IHCs with disorganized HB in ANKRD24 mutants. Is there a correlation between stereocilium loss and TRIOBP localization defects?
4. Figure S4M, N should be moved to main figures, and the stereocilium number quantified.
5. Line 116. The meaning of "full width at half-maximum mean" should be clearly explained for the broad readership of the journal.

Reviewer #2 (Comments to the Authors (Required)):

Summary

In this paper, the authors report on the role of ANKRD24 as a new factor that contributes to shaping the bundles of stereocilia that extend from the apical surface of sensory hair cells. ANKRD24 localizes to the stereocilium insertion point, wrapping around the core bundle rootlets in a ring link pattern. The authors use microscopy and biochemistry to make that case that ANKRD24 works with the known rootlet bundler TRIOBP-5 to maintain the mechanical stability and resilience of these structures. ANKRD24 is poorly characterized and this paper covers a lot of ground in terms of generating new insight, not only on molecular mechanisms, but also its contributions to hearing physiology. Overall, I think this is a great paper that is well-matched for the J Cell Biol. However, I do have some questions and suggestions for improving the strength of the story, especially with regard to the quantification of microscopy data.

General points

This paper is overwritten. I do not believe it should take 10 paragraphs to describe the localization of a molecule from light microscopy images (see description of Fig. 1). While I appreciate the careful descriptions provided by the authors, the readability of the paper as a whole is compromised with this approach. I hope the authors will consider this point when revising the paper.

Figs. 2, 3, 4 and others - The authors provide virtually no image quantification, which limits the confidence that the reader is able to invest in these data. This becomes especially important with the co-expression experiments and structure/function analysis in figures 3 and 4. In cases where the authors are comparing the localization/colocalization of ANKRD24 and TRIOBP, simple pixel by pixel correlation would be enough; these measurements are very easy to make on multiple images to generate values that could be compiled and subject to statistical testing.

Specific points

Line 33 - "Delicate mechanosensory cells require unusual cell biological specializations. "
Please delete this sentence.

Line 54 - "Stereocilia rootlets span this pivot point."

Span does not seem like the right word here as the core bundle extends through the insertion point down into the cytoplasm.

In a few different points in the narrative (pp. 4 and 8), the authors refer to a "clear channel" surrounding the rootlet, but to a naïve reader it's not clear what that is.

Line 167 - "Thus, ANKRD24 but not TRIOBP appears to be associated with membrane structures within HeLa cells" and Line 347 - "As with RAI14 (Wolf et al., 2019), the N-terminal 12 amino acids-predicted to be an amphipathic helix-helps direct ANKRD24 to the plasma membrane."

Based on the data shown in Fig. 4 I cannot agree with this conclusion. The idea that ANKRD24 might be binding to curved membrane at the insertion point is fascinating, but the images in Fig. 4G and H look identical to me. This is one place where quantitative analysis of protein localization becomes critical. If the authors want to make this claim, they must show a quantitative analysis that supports this point.

Related to the previous point - If ANKRD24 is somehow detecting membrane curvature at the insertion points, is it found in regions of positive curvature when exogenously expressed in culture cells? This did not appear to be the case in the HeLa cell experiments where ANKRD24 was mostly found on cytoplasm structures that labeled with lectin. Does ANKRD24 target to the base of filopodia, a region that demonstrates a comparable positive curvature, in these or other cultured cells?

Line 402 - "TRIOBP-5 deficient stereocilia of *Triobp* Δ 9- Δ 10 mice lacked ANKRD24 expression within the rootlets due to mislocalization of ANKRD24 (Fig. 8)."

Would one say that ANKRD24 is "expressed within the rootlets"? The authors confirmed that ANKRD24 was mislocalized, but I did not see data on reduced expression.

Line 410 - "ANKRD24 is recruited to the rootlet by TRIOBP-5, but the presence of ANKRD24 enables the proper maturation of the rootlet."

Although this is consistent with the observations in mice, the studies in HeLa cells suggest that ANKRD24 can also impact the distribution of TRIOBP-5.

Line 446 - "Without ANKRD24, early postnatal hair bundles become misshaped, like those of TRIOBP-5 deficient bundles, indicating the importance of ANKRD24 for anchoring each stereocilium in place as soon as rootlets begin to develop."

Indeed, it looks like the primary morphological phenotype in the ANKRD24 KO mouse is a failure to position stereocilia properly. Can the authors hazard a guess as to how/why linking rootlets into the cuticular plate constrains their organization into the typical U-shaped bundle? Are elements of the cuticular plate patterned in this shape?

Editor:

As you will see, both reviewers are overall positive regarding your study. We agree that for resubmission to JCB all requested quantifications are essential. However, while reviewer #1 has provided several interesting suggestions of points to investigate, as these are not essential for your main conclusions providing experimental data to address them is optional.

The reviewers' comments were quite helpful, especially their requests to quantify the cultured-cell data.

Reviewer #1:

In this manuscript by Krey et al., the authors report an essential role of the ankyrin-repeat protein ANKRD24 in structural stability of the hair cell stereocilia through extensive functional and biochemical analysis. Using super-resolution light microscopy and structure-function analysis in heterologous cells, they show that ANKRD24 localizes to the stereocilium insertion point and lower rootlet, interacts with the rootlet actin regulator TRIOBP and they mutually regulate each other's localization. ANKRD24 knockout mice have misshapen hair bundles (HB), progressive hearing loss and vulnerability to noise damage. Overall, this is a thorough and rigorous analysis that convincingly establishes ANKRD24 as a key structural component of the stereocilium rootlet crucial for HB stability and hearing.

There are several issues that require clarification and/or further interrogation, as detailed below:

1. It is not clear how ANKRD24 is localized to the stereocilium insertion point. There is no evidence that the N-terminal membrane-interacting region of ANKRD24 is sensitive to membrane curvature. Another possibility worth considering and investigating is that ANKRD24 interacts with specific lipids enriched in the previously defined "taper domain" (Zhao et al., 2012).

In Zhao et al. (2012), the lipids identified as being enriched in the taper region are gangliosides, which are sialic acid modified, ceramide-based glycosphingolipids that are likely exclusively found on the extracellular leaflet. ANKRD24 will bind to lipids on the intracellular leaflet. Although it is reasonable to suggest that there are particular lipids found at the insertion point given the high curvature of the membrane, we do not at present know what they are.

We certainly agree with the reviewer that the ANKRD24 membrane binding could be the result of binding to a unique lipid environment. Therefore, we added the following text to the Discussion: "Alternatively, ANKRD24 may be targeted to the taper by an unusual lipid composition (Zhao et al., 2012)."

2. Related to above, it is also possible that ANKRD24-membrane interaction is allosterically regulated by ANKRD24-TRIOBP interaction, as suggested by the predicted intramolecular interaction (Line 357). This should be discussed and investigated by more carefully examining ANKRD24 localization relative to cell membranes in TRIOBP5 mutants at multiple developmental stages, in addition to data shown in Figure 8P-S.

It is plausible that the ANKRD24-membrane interaction is allosterically regulated by the ANKRD24-TRIOBP interaction. However, the reviewer's experiment is a difficult one to carry out. We can see ANKRD24 so clearly in control hair cells because it is concentrated at 50-100 small points. However, ANKRD24 is difficult to visualize in the *Triobp* mutants because it is no longer concentrated at the insertions. While ANKRD24 may bind other membranes in these mutants, it is below the limit of detection.

Moreover, it would be informative to investigate the localization of ANKRD24- Δ ANK and Trunc2 using the injectoporation assay, to shed light on this potential regulation.

Our prediction would be that Trunc2 (with ANK repeats but lacking TRIOBP-binding domain) would not bind to rootlets and may or may not bind to the membrane at the insertion. ANKRD24- Δ ANK might still bind to rootlets, but not to the membrane—so it might be more uniformly distributed along the rootlet. Neither experiment would directly test the idea that the ANKRD24-membrane interaction is regulated allosterically by the ANKRD24-TRIOBP interaction, as a lack of binding to membranes could be the result of aberrant behavior of the recombinant protein.

These would be interesting experiments to do, especially as we further investigate ANKRD24 in the future. However, the injectoporation experiments would need to be done in *Ankrd24* KO cochlear explants and they cannot be done promptly—we would have to ship mice to the Zhao lab (who have the expertise in injectoporation). Because of the frequent low temperatures in Indianapolis, we might have to wait a few months to ship the mice. Then they would have them go through quarantine and be bred for these experiments—a few more months.

While these are important experiments for future studies of ANKRD24, we also agree with the editor that they are not essential to our main conclusions.

3. It appears that the area of the fonticulus in the ANKRD24 mutant hair cells is aberrantly enlarged (Figure 7P-W, more apparent in OHCs), suggesting possible "erosion" of the cuticular plate. This defect should be quantified and discussed.

Yes, the reviewer has noted a subtle phenotype that we think is related to polarity defects that are apparent in the *Ankrd24* KO hair cells. It is not so much that the fonticulus has widened, but rather instead that the connection of the cuticular plate to the cell borders has been altered. The cuticular plates themselves look relatively unchanged in the KOs compared to the hets, but there are more actin-free gaps between the cuticular plate and the cell border.

This phenotype is difficult to quantify but more importantly, will be investigated in our next ANKRD24 paper that will look at polarity (and the relationship of MYO7A and ANKRD24). We have added the following comment in the revised text to acknowledge the phenotype: "*We also noted increased gaps in actin labeling between the cuticular plate and the cell borders in Ankrd24^{KO/KO} hair cells, especially OHCs (Fig. 7V-W).*"

4. The warped HB morphology defects in ANKRD24 mutants are not seen in TRIOBP mutants (Kitajiri et al., 2010), suggesting that ANKRD24 and TRIOBP have separate functions during HB morphogenesis. This should be discussed.

The warped bundle phenotype of TRIOBP mutants was shown in supplemental figure 1 of Kitajiri et al. (2010), although the phenotype was less prominent than in *Ankrd24* mutants. We added a more compelling example of warped bundle phenotype in *Triobp-5* knockout hair cells to our Supplemental Figure S5AC, and added the following comment to the Results: "*The warped hair-bundle phenotype was present in these hair cells, albeit less prominent than in Ankrd24^{KO/KO} mice (Fig. S5AC).*" In the Discussion, we noted: "*Bundles of hair cells lacking TRIOBP are less misshaped than those in Ankrd24^{KO/KO} mice, suggesting that ANKRD24 has a role beyond simply connecting TRIOBP-5 to the membrane.*"

5. Given the progressive hearing loss and susceptibility to noise damage in ANKRD24 mutants, HB morphology should be assessed by SEM in older adults and following noise exposure.

Stereocilia imaging with phalloidin staining and super-resolution microscopy gives sufficient image quality to assess the consequences of aging and noise damage. We have added phalloidin-stained images of cochlea examined at older time points (Fig. 5J-K) and following noise damage (Fig. 9J-M).

Specific comments:

1. The max-intensity projection images shown in Figures 2-4 do not convincingly demonstrate co-localization. Quantifications of Pearson's coefficient should be performed using single-plane images.

Co-localization is perhaps not the best term, especially with such high resolution imaging—co-aggregation might be more apt. From the hair cell data it is clear that that ANKRD24 and TRIOBP do not completely overlap but rather create structures that are in close proximity to each other, perhaps intermeshed or intertwined. At sufficiently high resolution, we can distinguish ANKRD24 and TRIOBP, so a Pearson's coefficient analysis may not be ideal for revealing the overlap.

Nevertheless, we agree with the reviewer that we need to quantify the overlap between complexes, which allows us to conclude which proteins and domains are significant for those interactions. In the revised submission, we used a surface-rendering approach with the program Imaris (<https://imaris.oxinst.com/learning/view/article/imaris-9-surfaces-rendering-technology-for-large-images>) to identify volumes within the cell that correspond to ANKRD24 and TRIOBP aggregates, then

determined the overlap between those volumes. We quantified these overlaps in two different ways, which each show similar results.

Fig. 3G and H showed that the overlap of ANKRD24 with TRIOBP-1 and TRIOBP-5 were statistically similar and large, i.e., they interact, while the overlap of ANKRD24 with TRIOBP-4 was significantly less than the overlap with TRIOBP-5. As with the other data in Fig. 3, these data suggested that the C-terminal coiled-coiled regions of TRIOBP are the predominant domains interacting with ANKRD24.

Results in Fig. 4P-Q were even more clear. Examining TRIOBP-5 aggregates co-expressed with those of ANKRD24 deletion and truncation constructs showed that deletion of the ANKRD24 C-terminus in Trunc2, Trunc3, and Trunc5 led to a loss of interaction with TRIOBP-5.

We also quantified membrane localization of ANKRD24-GFP (interacts with membrane) and GFP-TRIOBP-1 (does not), using WGA to mark the plasma membrane in **unpermeabilized** cells. To quantify membrane localization, we used profiles through the membrane, measuring the relative intensity of the GFP signal at the peak of the WGA signal compared to the average inside the cell (Fig. 2J). These data showed compellingly that ANKRD24-GFP interacts with the membrane but that GFP-TRIOBP-5 does not.

Unfortunately, the WGA plasma membrane signal is not sharp enough after **permeabilization** (used for detecting dual-transfected HA- and Myc-tagged proteins) to allow relative quantitation of the plasma membrane targeting of the deletion constructs in Fig. 4.

2. The VsEP (and DPOAE) data in Figure 5 were based on small n's and not yet significant ($p=0.068$). Suggest either increasing n's to reach statistical significance or remove from the main Figure.

For the behavioral testing, we phenotyped the number of animals that we had available. Although there was a clearly discernible phenotype for the auditory system, there was no striking phenotype for the vestibular system. For example, the DPOAE results were significant at a very low p value despite the small n.

We have chosen to leave the VsEP data in because our point was to show that there was no statistically significant differences. We removed the clause that read "...although there was a trend towards threshold elevation for *Ankrd24*^{KO/KO} mice."

3. Figure 6J showed IHCs with normal HB morphology, whereas Figure 8G showed IHCs with disorganized HB in ANKRD24 mutants. Is there a correlation between stereocilium loss and TRIOBP localization defects?

We see what the reviewer is pointing to. However, in Fig. 6J, we display stereocilia surfaces rendered by Imaris. By contrast, in Fig. 8G-H, we show the actual fluorescence images from a stack, projected to get these views. Thus, it is an apples and oranges comparison, as the surfaces appear somewhat different from the phalloidin stained images.

We replaced the images in Fig. 6 with more representative ones. We have not noted a correlation between stereocilia loss and TRIOBP localization defects.

4. Figure S4M, N should be moved to main figures, and the stereocilium number quantified.

We have added these panels and the quantitation to Fig. 9.

5. Line 116. The meaning of "full width at half-maximum **mean**" should be clearly explained for the broad readership of the journal.

We did not use the term "full width at half-maximum **mean**", just "full width at half-maximum." To explain that term more completely, we have added the following explanatory sentence to the section in Methods about fluorescence microscopy: "The full-width at half-maximum (FWHM) is the side-to-side width of the Gaussian function measured at 50% of the peak of the function."

Reviewer #2:

Summary

In this paper, the authors report on the role of ANKRD24 as a new factor that contributes to shaping the bundles of stereocilia that extend from the apical surface of sensory hair cells. ANKRD24 localizes to the stereocilium insertion point, wrapping around the core bundle rootlets in a ring link pattern. The authors use microscopy and biochemistry to make that case that ANKRD24 works with the known rootlet bundler TRIOBP-5 to maintain the mechanical stability and resilience of these structures. ANKRD24 is poorly characterized, and this paper covers a lot of ground in terms of generating new insight, not only on molecular mechanisms, but also its contributions to hearing physiology. Overall, I think this is a great paper that is well-matched for the J Cell Biol. However, I do have some questions and suggestions for improving the strength of the story, especially about the quantification of microscopy data.

General points

This paper is overwritten. I do not believe it should take 10 paragraphs to describe the localization of a molecule from light microscopy images (see description of Fig. 1). While I appreciate the careful descriptions provided by the authors, the readability of the paper as a whole is compromised with this approach. I hope the authors will consider this point when revising the paper.

We have revised and condensed the paper, which was also necessary to try to meet the journal's length requirement.

Figs. 2, 3, 4 and others - The authors provide virtually no image quantification, which limits the confidence that the reader is able to invest in these data. This becomes especially important with the co-expression experiments and structure/function analysis in figures 3 and 4. In cases where the authors are comparing the localization/colocalization of ANKRD24 and TRIOBP, simple pixel by pixel correlation would be enough; these measurements are very easy to make on multiple images to generate values that could be compiled and subject to statistical testing.

Yes, as also pointed out by reviewer #1, quantitation of the imaging in HeLa cells was lacking. Additional quantification of HeLa cell experiments was done and is provided in the revised manuscript as requested.

Specific points

Line 33 - "Delicate mechanosensory cells require unusual cell biological specializations."

Please delete this sentence.

Deleted.

Line 54 - "Stereocilia rootlets span this pivot point."

Span does not seem like the right word here as the core bundle extends through the insertion point down into the cytoplasm.

Changed to "*Stereocilia rootlets traverse this pivot point.*"

In a few different points in the narrative (pp. 4 and 8), the authors refer to a "clear channel" surrounding the rootlet, but to a naïve reader it's not clear what that is.

Page 5: changed to "...allowed visualization of the actin-free channel surrounding the rootlet..."

Page 8: made several changes, removing "clear channel."

Line 167 - "Thus, ANKRD24 but not TRIOBP appears to be associated with membrane structures within HeLa cells" and Line 347 - "As with RAI14 (Wolf et al., 2019), the N-terminal 12 amino acids-predicted to be an amphipathic helix-helps direct ANKRD24 to the plasma membrane." Based on the data shown in Fig. 4 I cannot agree with this conclusion. The idea that ANKRD24 might be binding to curved membrane at the insertion point is fascinating, but the images in Fig. 4G and H look identical to me. This is one place where quantitative analysis of protein localization becomes critical. If the authors want to make this claim, they must show a quantitative analysis that supports this point.

Comparison of Fig. 4E (ΔN) to Fig. 4D (WT) is a better comparison of ΔN to WT than Fig. 4H (ΔN) to Fig. 4G (WT), as the latter panels were chosen to highlight the intracellular aggregates. Quantitation of

membrane localization in the experiments of Fig. 4, which used tagged constructs that required permeabilization, is less accurate than the experiments that used GFP constructs and no permeabilization (e.g., Fig. 2). WGA labeling in permeabilized cells did not mark the plasma membrane as clearly in permeabilized cells as in unpermeabilized cells.

That said, the reviewer is correct in noting that ΔN still has some plasma membrane labeling. We toned down our conclusions. In the Results: "*Plasma membrane localization of ΔN was reduced in HeLa cells, although intracellular aggregates still contained membrane and actin (Fig. 4D-E, H; Fig. S3I).*" In the Discussion: "*As with RAI14 (Wolf et al., 2019), the N-terminal 12 amino acids increases targeting of ANKRD24 to the plasma membrane.*" (Bold added to indicate significant changes.)

Related to the previous point - If ANKRD24 is somehow detecting membrane curvature at the insertion points, is it found in regions of positive curvature when exogenously expressed in culture cells? This did not appear to be the case in the HeLa cell experiments where ANKRD24 was mostly found on cytoplasm structures that labeled with lectin. Does ANKRD24 target to the base of filopodia, a region that demonstrates a comparable positive curvature, in these or other cultured cells?

We had already pointed out that ANKRD24 (and TRIOBP) targeted to the base of filopodia in cultured cells: "*Co-expressed ANKRD24 and either TRIOBP-1 or TRIOBP-5 were also co-localized at filopodia bases in HeLa cells (Fig. 3F-G).*" In the new version, we dropped the TRIOBP-1 profile image to allow space in the figure for the quantitation, so the sentence now reads as follows: "*Co-expressed ANKRD24 and TRIOBP-5 were co-localized at filopodia bases in HeLa cells (Fig. 3F).*"

Although we included the panels showing only aggregates, ANKRD24 was not "*mostly found on cytoplasm structures that labeled with lectin.*" We increased the gain for the ANKRD24 channel in Fig. 2F to make this point more clear; ANKRD24-GFP localizes not just to the cytoplasm structures, but also along the plasma membrane. Anecdotally, ANKRD24 localization was more prominent at larger areas of positive curvature at the plasma membrane, but we do not have a simple way to conclusively demonstrate that.

Line 402 - "TRIOBP-5 deficient stereocilia of *Triobp Δ ex9- Δ ex10* mice lacked ANKRD24 expression within the rootlets due to mislocalization of ANKRD24 (Fig. 8)." Would one say that ANKRD24 is "expressed within the rootlets"? The authors confirmed that ANKRD24 was mislocalized, but I did not see data on reduced expression.

This line was deleted.

Line 410 - "ANKRD24 is recruited to the rootlet by TRIOBP-5, but the presence of ANKRD24 enables the proper maturation of the rootlet." Although this is consistent with the observations in mice, the studies in HeLa cells suggest that ANKRD24 can also impact the distribution of TRIOBP-5.

We have modified the sentence according to the reviewer's comment. The sentence now reads "*Although ANKRD24 is recruited to the rootlet by TRIOBP-5, the presence of ANKRD24 enables the proper maturation of the rootlet.*"

Line 446 - "Without ANKRD24, early postnatal hair bundles become misshaped, like those of TRIOBP-5 deficient bundles, indicating the importance of ANKRD24 for anchoring each stereocilium in place as soon as rootlets begin to develop." Indeed, it looks like the primary morphological phenotype in the ANKRD24 KO mouse is a failure to position stereocilia properly. Can the authors hazard a guess as to how/why linking rootlets into the cuticular plate constrains their organization into the typical U-shaped bundle? Are elements of the cuticular plate patterned in this shape?

The reviewer is quite right that a significant morphological phenotype is the lack of proper stereocilia positioning. Interestingly, spacing between adjacent stereocilia seem to be normal in *Ankrd24* KOs, but longer-range organization is compromised. We are presently preparing a separate manuscript, also examining the relationship of ANKRD24 to MYO7A (see Morgan et al., 2016), which will be a follow-up to the Etournay et al 2010 Development paper from the Petit lab.

January 13, 2022

RE: JCB Manuscript #202109134R

Dr. Peter Barr-Gillespie
Oregon Health & Science University
Oregon Hearing Research Center
L335A
3181 SW Sam Jackson Pk Rd
Portland, OR 97239

Dear Dr. Barr-Gillespie:

Thank you for submitting your revised manuscript entitled "ANKRD24 organizes TRIOBP to reinforce stereocilia insertion points". We would be happy to publish your paper in JCB pending final revisions necessary to meet our formatting guidelines (see details below).

A. MANUSCRIPT ORGANIZATION AND FORMATTING:

- 1) Text limits: Character count for Articles is < 40,000, not including spaces. Count includes abstract, introduction, results, discussion, and acknowledgments. Count does not include title page, figure legends, materials and methods, references, tables, or supplemental legends.
- 2) Figures limits: Articles may have up to 10 main text figures.
- 3) Figure formatting: Scale bars must be present on all microscopy images, including inset magnifications. Molecular weight or nucleic acid size markers must be included on all gel electrophoresis.
- 4) Statistical analysis: Error bars on graphic representations of numerical data must be clearly described in the figure legend. The number of independent data points (n) represented in a graph must be indicated in the legend. Statistical methods should be explained in full in the materials and methods. For figures presenting pooled data the statistical measure should be defined in the figure legends. Please also be sure to indicate the statistical tests used in each of your experiments (either in the figure legend itself or in a separate methods section) as well as the parameters of the test (for example, if you ran a t-test, please indicate if it was one- or two-sided, etc.). Also, if you used parametric tests, please indicate if the data distribution was tested for normality (and if so, how). If not, you must state something to the effect that "Data distribution was assumed to be normal but this was not formally tested."
- 5) Abstract and title: The abstract should be no longer than 160 words and should communicate the significance of the paper for a general audience. The title should be less than 100 characters including spaces. Make the title concise but accessible to a general readership.
- 6) Materials and methods: Should be comprehensive and not simply reference a previous publication for details on how an experiment was performed. Please provide full descriptions in the text for readers who may not have access to referenced manuscripts.
- 7) Please be sure to provide the sequences for all of your primers/oligos and RNAi constructs in the materials and methods. You must also indicate in the methods the source, species, and catalog numbers (where appropriate) for all of your antibodies. Please also indicate the acquisition and quantification methods for immunoblotting/western blots.
- 8) Microscope image acquisition: The following information must be provided about the acquisition and processing of images:
 - a. Make and model of microscope
 - b. Type, magnification, and numerical aperture of the objective lenses
 - c. Temperature
 - d. Imaging medium
 - e. Fluorochromes
 - f. Camera make and model

g. Acquisition software

h. Any software used for image processing subsequent to data acquisition. Please include details and types of operations involved (e.g., type of deconvolution, 3D reconstitutions, surface or volume rendering, gamma adjustments, etc.).

10) Supplemental materials: There are strict limits on the allowable amount of supplemental data. Articles may have up to 5 supplemental figures. Please also note that tables, like figures, should be provided as individual, editable files. A summary of all supplemental material should appear at the end of the Materials and methods section.

13) ORCID IDs: ORCID IDs are unique identifiers allowing researchers to create a record of their various scholarly contributions in a single place. At resubmission of your final files, please consider providing an ORCID ID for as many contributing authors as possible.

Please note that JCB now requires authors to submit Source Data used to generate figures containing gels and Western blots with all revised manuscripts. This Source Data consists of fully uncropped and unprocessed images for each gel/blot displayed in the main and supplemental figures. Since your paper includes cropped gel and/or blot images, please be sure to provide one Source Data file for each figure that contains gels and/or blots along with your revised manuscript files. File names for Source Data figures should be alphanumeric without any spaces or special characters (i.e., SourceDataF#, where F# refers to the associated main figure number or SourceDataFS# for those associated with Supplementary figures). The lanes of the gels/blots should be labeled as they are in the associated figure, the place where cropping was applied should be marked (with a box), and molecular weight/size standards should be labeled wherever possible.

B. FINAL FILES:

Additionally, JCB encourages authors to submit a short video summary of their work. These videos are intended to convey the main messages of the study to a non-specialist, scientific audience. Think of them as an extended version of your abstract, or a short poster presentation. We encourage first authors to present the results to increase their visibility. The videos will be displayed with your manuscript on our website and will be shared on social media to promote your work. For more detailed guidelines and tips on preparing your video, please visit <https://rupress.org/jcb/pages/submission-guidelines#videoSummaries>.

Thank you for this interesting contribution, we look forward to publishing your paper in Journal of Cell Biology.

Sincerely,

John Wallingford, PhD
Monitoring Editor

Andrea L. Marat, PhD
Senior Scientific Editor

Journal of Cell Biology